# Co-Generative De Novo Functional Protein Design

Xinrui Chen [1 2]  Yizhen Luo [1 2]  Siqi Fan [1]  Zaiqing Nie[† 1 3]

## Abstract

*De novo* functional protein design aims to gener-
ate protein sequences that realize specified bio-
chemical functions without relying on evolution-
ary templates, enabling broad applications in
biotechnology and medicine. Existing approaches
adopt either direct function-to-sequence mapping
or decoupled structure-sequence generation strate-
gies but often fail to achieve functionality and
foldability simultaneously. To address this, we
propose **CodeFP**, a **Co**-generative protein lan-
guage model for *de novo* **F**unctional **P**rotein de-
sign that simultaneously decodes sequence and
structure tokens, thereby enabling superior simul-
taneous realization of functionality and foldability.
CodeFP utilizes functional local structures to en-
rich functional semantic encodings, overcoming
the suboptimal translation of flat encodings into
structure tokens, while introducing auxiliary func-
tional supervision to alleviate training ambigu-
ity stemming from the one-to-many structure-to-
token mapping. Extensive experiments show that
CodeFP consistently achieves average improve-
ments of 6.1% in functional consistency and 3.2%
in foldability over the strongest baseline.

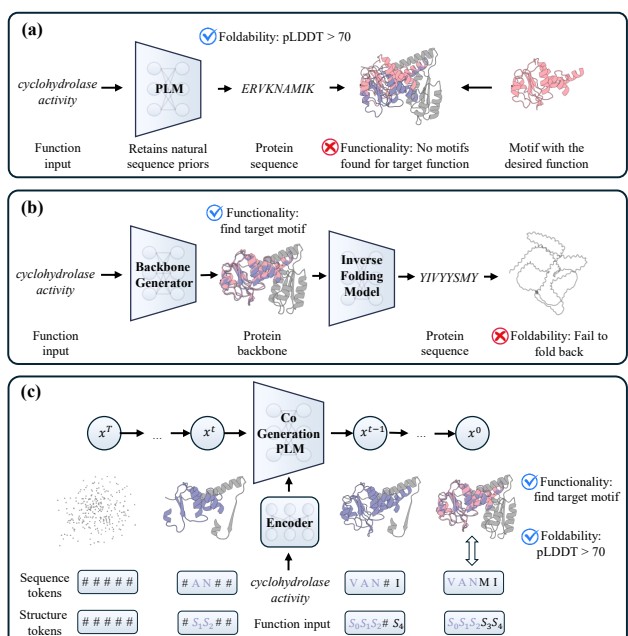

*Figure 1.* **Motivation of CodeFP. (a)** One-step generation (limited
functional control); **(b)** Two-step generation (unreliable foldabil-
ity); **(c)** CodeFP (joint sequence-structure decoding). By iteratively
generating both sequence and structure tokens, CodeFP ensures
that the generated proteins possess valid folds while retaining criti-
cal functionality.

## 1. Introduction

Functional protein design aims to engineer novel sequences
with tailored biological functions, enabling the diverse cre-
ation of enzymes with enhanced catalytic efficiency (Austin
et al., 2018; Khersonsky et al., 2018; Munsamy et al., 2022),
therapeutic proteins with low toxicity (Marshall et al., 2003;
Chun et al., 2025), and antibodies with improved binding
specificity (Leaver-Fay et al., 2016). Recently, *de novo*
functional protein design has attracted increasing interest in

biological research (Yeh et al., 2023; Kortemme, 2024). Un-
like traditional approaches that optimize existing wild-type
proteins via directed evolution (Stemmer, 1994; Savile et al.,
2010; Yang et al., 2019a), *de novo* design operates beyond
the space of naturally occurring sequences, thereby jumping
out of local fitness optima and enabling novel combinations
of multiple functions within a single protein.

Recent advances in machine learning have attempted to ad-
dress *de novo* functional protein design as a conditional gen-
eration task, employing Gene Ontology (GO) terms (Ash-
burner et al., 2000) or natural language to model the desired
function. These methods could be generally categorized
as follows: (1) One-step generation (Madani et al., 2020;
Munsamy et al., 2022; Yin et al., 2025; Liu et al., 2025)
leverages autoregressive or diffusion-based pre-trained Pro-
tein Language Models (PLMs) to map functional conditions
directly to amino acid sequences. (2) Two-step generation

[†]Corresponding author. [1]Institute for AI Industry Research
(AIR), Tsinghua University [2]Department of Computer Science and
Technology, Tsinghua University [3]PharMolix Inc. Correspondence
to: Xinrui Chen <cxr21@mails.tsinghua.edu.cn>, Zaiqing Nie
<zaiqing@air.tsinghua.edu.cn>.

*Proceedings of the $43^{rd}$ International Conference on Machine
Learning*, Seoul, South Korea. PMLR 306, 2026. Copyright 2026
by the author(s).

(Watson et al., 2023; Ingraham et al., 2023; Dai et al., 2024) incorporates structure as an explicit intermediate modality. Specifically, these models first generate a backbone conditioned on the desired function and then derive the sequence via inverse folding (Dauparas et al., 2022).

However, due to the intricate coupling among protein sequence, structure, and function, these methods often struggle to generate proteins that simultaneously exhibit **foldability**, *i.e.,* the sequence should fold into a stable and well-defined three-dimensional structure, and **functionality**, *i.e.,* the generated protein should exhibit the desired functions. Specifically, (1) One-step generation, while promoting robust foldability by inheriting natural sequence priors from pre-trained PLMs, often results in degraded functionality due to the diverse sequence realizations underlying a given function that complicates learning, as illustrated in Fig. 1(a). (2) Two-step generation, while explicitly modeling structure to ground function, leads to suboptimal foldability as it neglects sequence constraints during backbone generation, yielding geometries that are incompatible with folding back into a natural sequence, as illustrated in Fig. 1(b).

In light of recent advances in co-generative PLMs (Wang et al., 2024b; Hayes et al., 2025; Yang et al., 2025), we introduce **CodeFP**, a novel **Co**-generative PLM framework for **de** novo **Fun**ctional protein design. CodeFP quantizes local structures for each amino acid into discrete tokens and models them jointly with the protein sequence. During generation, the two modalities are decoded in an interleaved manner, thereby enhancing function modeling via structural integration and ensuring foldability by incorporating sequence constraints, as illustrated in Fig. 1(c).

Notably, we observe two technical challenges when extending this strategy to *de novo* functional protein design. First, following prior work (Yin et al., 2025; Dai et al., 2024) that encodes functions with one-hot vectors or natural language embeddings and translates them into structure tokens is suboptimal, as it overlooks the hierarchical structure of protein functions and the intricate connections between functions and proteins. Inspired by motif scaffolding (Wang et al., 2022), we retrieve and encode functional structural motifs. These representations are aggregated by functional category and subsequently integrated via cross-attention to condition the generative process, enhancing the translation from function terms to proteins. Second, since structural tokenization is sensitive to global topology, functional motifs exhibit diverse realizations in structural token sequences. However, the training objective of discrete diffusion treats them as competing modes, leading to ambiguity. To mitigate this, we apply a functional prediction head to the continuous hidden states of generated local structural motifs as an auxiliary training signal, facilitating function-conditioned learning.

Extensive experiments demonstrate that CodeFP achieves superior functionality and foldability compared to state-of-the-art methods. Quantitatively, it yields a 7.6% gain in functional F1-Macro and improves the foldability success rate (pLDDT $> 70$) by 5.2% over the strongest baseline. Notably, a 9.1% improvement in F1-Macro in the out-of-distribution (OOD) test set indicates that CodeFP possesses superior generalization capabilities for unseen functional combinations. Our contributions are summarized as follows:

- We propose CodeFP, a co-generative PLM framework for *de novo* functional protein design that effectively satisfies both functionality and foldability.

- We aggregate function-specific motifs to capture stronger function semantics, while introducing an auxiliary training signal to mitigate ambiguity arising from structure discretization.

- CodeFP achieves the best joint performance in functionality and foldability among all compared methods, establishing a new state-of-the-art in *de novo* functional protein design.

## 2. Related Work

**Protein Generative Models.** Generative approaches for protein design can be categorized into three paradigms based on their modeling modalities. (1) Sequence generation models the probability distribution of amino acids to capture evolutionary patterns. Early approaches, including Prot-GPT2 (Ferruz et al., 2022) and Prollama (Lv et al., 2025), employ autoregressive language models to generate protein sequences. In contrast, recent discrete diffusion models like DPLM (Wang et al., 2024a) and EvoDiff (Alamdari et al., 2023) formulate protein generation as an iterative denoising process. (2) Structure generation focuses on constructing valid backbone geometries. These methods typically model continuous 3D backbone geometries using diffusion or flow-matching frameworks, including RFdiffusion (Watson et al., 2023) and FoldFlow (Bose et al., 2023), whereas approaches such as SLM (Lu et al., 2024) generate autoregressively over discretized structural tokens. (3) Co-generation accommodates these modalities, enforcing sequence–structure consistency during generation. Representative approaches couple both modalities using multi-modal flow matching, as seen in MultiFlow (Campbell et al., 2024), or employ dual-channel discrete diffusion, utilized by ESM3 (Hayes et al., 2025), and DPLM-2 (Wang et al., 2024b). Building on this emerging paradigm, CodeFP leverages FSR and LSFS to further align these modalities with functional constraints, effectively extending co-generation PLMs to the task of *de novo* functional protein design.

**Functional Protein Design.** Functional protein design aims to generate protein sequences with specific biological func-

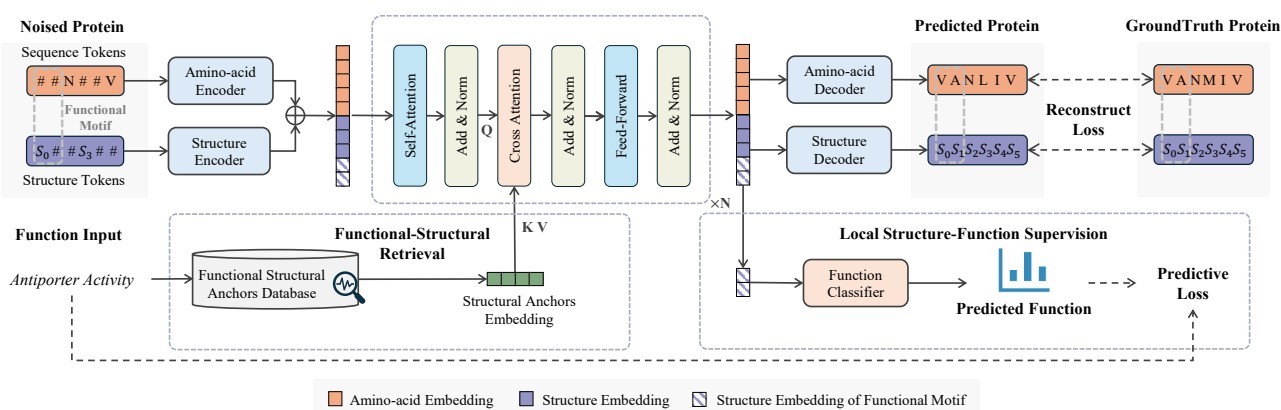

*Figure 2.* **The overall architecture of CodeFP.** CodeFP facilitates *de novo* functional protein design through a co-generation process. Given a function prompt, the Functional-Structural Retrieval module retrieves representative structural motifs as informative priors. These priors guide the Co-generation Transformer to iteratively reconstruct sequence and structure tokens via cross-attention. In parallel, the Local Structure-Function Supervision module provides auxiliary training signals by classifying the embedding of generated functional local structures. The entire system is trained by minimizing a joint objective of reconstruction and predictive losses.

tions. Early evolution-based methods navigated fitness landscapes to optimize sequences derived from natural variants (Yang et al., 2019b). In contrast, recent approaches leverage *de novo* generative models to design novel proteins, generally following either a one-step or two-step paradigm. (1) One-step approaches focus on direct sequence generation conditioned on function. ProteoGAN (Kucera et al., 2022) utilizes GANs to model label-sequence relationships, while ProGen2 (Madani et al., 2020) and ZymCTRL (Munsamy et al., 2022) leverage autoregressive PLMs for functional steering. Recently, CFP-Gen (Yin et al., 2025) introduced discrete diffusion to satisfy multiple constraints. (2) Two-step approaches prioritize backbone generation: Chroma (Ingraham et al., 2023) and ProDiT (Jing et al., 2025) generate continuous coordinates via diffusion or flow-matching, whereas Pinal (Dai et al., 2024) predicts discrete structural tokens before amino acid design. In this work, CodeFP simultaneously generates sequence and structure, integrating the strengths of one-step and two-step approaches.

## 3. Method

In this section, we present the model architecture of CodeFP that facilitates the simultaneous achievement of functionality and foldability. We begin by formalizing the problem and introducing the co-generation framework in Section 3.1. Next, Section 3.2 details the retrieval module, which improves the suboptimal translation of flat semantic encodings. Finally, Section 3.3 describes the auxiliary supervision, which alleviates the training ambiguity caused by discretization.

### 3.1. Generating functional proteins with co-generation

**Problem Formulation.** We formulate *de novo* functional protein design by representing a protein as $\mathcal{P} = (\mathbf{s})$, where $\mathbf{s} = [s_1, \ldots, s_L]$ is an amino acid sequence of length $L$ and each residue $s_i \in \mathcal{V}_{\text{seq}}$ is drawn from the 20 standard amino acids, and specifying its target function using GO molecular function terms $c_{\text{GO}}$, which provide hierarchical labels that support general functional descriptions. The objective is to model the conditional distribution $p(\mathcal{P} \mid c_{\text{GO}})$.

**Structure Quantization for Discrete Diffusion.** Following DPLM-2, we extend the protein defination to $\mathcal{P} = (\mathbf{s}, \mathbf{x})$, where $\mathbf{x} \in \mathbb{R}^{L \times 4 \times 3}$ denotes the backbone atom coordinates (N, C$\alpha$, C, O). Then a LFQ-based (Yu et al., 2023) vector-quantized structure tokenizer maps $\mathbf{x}$ to discrete structure tokens $\mathbf{z} = [z_1, \ldots, z_L]$ by capturing local structure contexts of each amino acid. Here, $z_i \in \{0, \ldots, |\mathcal{V}_{\text{struct}}| - 1\}$ where $\mathcal{V}_{\text{struct}}$ is a fixed-size vocabulary set. This results in a unified discrete representation $\mathcal{P}_{disc} = (\mathbf{s}, \mathbf{z})$.

**Forward Process: Multimodal Absorbing Diffusion.** We model the joint distribution of $\mathcal{P}_{\text{disc}}$ using discrete diffusion (Austin et al., 2021) with an absorbing corruption process. At each step, tokens are progressively replaced by a modality-specific mask token [MASK]. Let $\mathbf{u}^{(t)} = (\mathbf{s}^{(t)}, \mathbf{z}^{(t)})$ denote the state at diffusion step $t \in \{0, \ldots, T\}$, where $\mathbf{u}^{(0)}$ is the clean data and $\mathbf{u}^{(T)}$ approaches a fully masked noise distribution. The forward process is a Markov chain $q(\mathbf{u}^{(t)} \mid \mathbf{u}^{(t-1)})$ with independent transitions across positions and modalities.

For any token $u \in s, z$, the transition is defined as

$$q(u^{(t)} \mid u^{(t-1)}) = \text{Cat}\left(u^{(t)}; u^{(t-1)}\mathbf{Q}_t\right), \quad (1)$$

where the absorbing transition matrix is

$$\mathbf{Q}_t = \text{diag}(1 - \beta_t) + \beta_t \cdot \mathbf{1}[\text{MASK}], \qquad (2)$$

Here, $\beta_t$ controls the corruption rate, and $\mathbf{1}[\text{MASK}]$ assigns all probability mass to the absorbing mask state.

**Reverse Denoising with Functional Conditioning.** The generative process reconstructs the clean protein $\mathbf{u}^{(0)}$ from the corrupted state $\mathbf{u}^{(t)}$ by reversing the diffusion trajectory conditioned on $\mathcal{C} = \{\mathbf{C}_{GO}\}$, which encodes functional semantics of GO terms. The reverse transition is approximated by marginalizing over the predicted clean state:

$$p_\theta(\mathbf{u}^{(t-1)}|\mathbf{u}^{(t)}, \mathcal{C}) \propto$$
$$\sum_{\tilde{\mathbf{u}}^{(0)}} q(\mathbf{u}^{(t-1)}|\mathbf{u}^{(t)}, \tilde{\mathbf{u}}^{(0)}) p_\theta(\tilde{\mathbf{u}}^{(0)}|\mathbf{u}^{(t)}, \mathcal{C}), \qquad (3)$$

where $p_\theta(\cdot|\mathbf{u}^{(t)}, \mathcal{C})$ denotes the neural network prediction. By sustaining dense mutual interaction at every denoising step, our iterative decoding strategy ensures that structural generation is tightly constrained by sequence constraints. Simultaneously, this progressive refinement grants the structural topology sufficient flexibility to extensively explore the geometric space for functional alignment.

**Optimization Objective.** The generative objective $\mathcal{L}_{\text{gen}}$ minimizes the variational lower bound for the joint distribution, which reduces to a weighted sum of negative log-likelihoods. Independent time steps $t_s$ and $t_z$ are sampled for sequence and structure modalities, respectively. The loss is formulated as:

$$\mathcal{L}_{\text{gen}} = \mathbb{E}_{q(\mathbf{u}^{(0)})}\Bigg[ \sum_{i=1}^{L} \Big( \lambda(t_s)b_i(t_s)\mathcal{L}_{\text{seq}}^{(i)}$$
$$+ \lambda(t_z)b_i(t_z)\mathcal{L}_{\text{struct}}^{(i)} \Big) \Bigg], \qquad (4)$$

where $b_i(t) \in \{0, 1\}$ indicates whether the token at position $i$ is masked at time $t$, and $\mathcal{L}^{(i)}$ represents the negative log-likelihood of the reconstruction.

### 3.2. Functional-Structural Retrieval

Protein functional semantics exhibit deep dependencies on both sequence and structure. Existing methods (Dai et al., 2024; Yin et al., 2025), which rely on one-hot encodings or textual embeddings, suffer from two critical limitations. First, they neglect the hierarchical context of biological functions, such as ATPase activity, which often necessitate capabilities like electron transport. Second, they suffer from geometric decoupling, ignoring the physical reality that functions like ligand binding or enzymatic catalysis are instantiated by specific structural motifs. To address these limitations, we ground functional labels in their physical

manifestations. As illustrated in Fig. 2, our method proceeds in two phases: constructing a retrieval database of functional structural motifs, and injecting these priors via cross-attention.

**Construction of Functional Structural Representation.** We construct a retrieval database $\mathcal{M}$ from the training set, mapping each GO term to a continuous embedding representing its geometric realization. This process comprises two steps: Representation Encoding and motif Aggregation.

*Representation Encoding.* Since specific biological functions are governed by local structural motifs rather than the global fold, accurately modeling function requires isolating its geometric instantiation. To achieve this, we utilize the pre-computed domain terms provided in our training set, derived using InterProScan (IPS) (Jones et al., 2014). Based on these terms, we extract the local backbone coordinates $\mathbf{x}_{local}$ corresponding to each protein-GO pair. To translate this geometry into a functional semantic space, we encode $\mathbf{x}_{local}$ using the frozen DPLM-2 encoder—ensuring alignment with our CodeFP backbone. Specifically, the coordinates are discretized via LFQ and processed to extract the `[CLS]` representation $\mathbf{e}_{i,j}$. This resulting embedding effectively captures the intrinsic dependency between the function and its underlying local structure.

*Motif Aggregation.* A GO term $y$ may be associated with diverse proteins, each carrying evolutionary specificities unrelated to the core function. To distill the essential geometric signature of the function and inject an inductive bias for hierarchical protein function, we compute the structural motif $\mathbf{c}_y$ by averaging all local structure embeddings $\mathbf{e}_{i,y}$ associated with label $y$ (i.e., $\mathbf{c}_y = \text{Mean}(\{\mathbf{e}_{i,y} \mid (P_i, y) \in \mathcal{S}_y\})$). Crucially, aggregation preserves the hierarchical structure of function, since the aggregate representation of a parent function naturally encompasses its child nodes. These centroids serve as hierarchically aware structural motifs, forming our retrieval database $\mathcal{M} = \{(y, \mathbf{c}_y)\}_{y \in \mathcal{Y}}$.

**Injection via Cross-Attention.** During both training and inference, we inject these structural motifs into the co-generation process. Given a set of input GO labels $\mathcal{Y}_{in}$, we retrieve their corresponding structural motifs $\mathbf{C} = \{\mathbf{c}_y \mid y \in \mathcal{Y}_{in}\}$. These motifs are then fused into the model representation via cross-attention layers, augmenting the conditioning set of the reverse denoising process from $\mathcal{C} = \{\mathbf{C}_{GO}\}$ to $\mathcal{C} = \{\mathbf{C}_{GO}, \mathbf{C}\}$. Let $\mathbf{H}^{(l)}$ denote the hidden states of the sequence and structure tokens at layer $l$. The injection is formulated as:

$$\mathbf{H}^{(l)'} = \mathbf{H}^{(l)} + \text{CrossAttn}(\mathbf{Q} = \mathbf{H}^{(l)}, \mathbf{K} = \mathbf{C}, \mathbf{V} = \mathbf{C}), \qquad (5)$$

where the generated tokens (Query) attend to the retrieved structural motifs (Key/Value). By grounding hierarchical functional knowledge in a structural perspective, we intro-

*Table 1.* **Main results on GO-conditioned protein design.** We evaluate functionality using the DeepGO-SE classifier. ↑ indicates higher is better, ↓ indicates lower is better. The best results among generative models are highlighted in **bold**, and the second best are underlined. Positive Control represents real proteins from the test set.

| Category | Model | F1-Micro (↑) | F1-Macro (↑) | AUPR (↑) | AUC-ROC (↑) | MRR (↑) | MMD (↓) | MMD-G (↓) |
|---|---|---|---|---|---|---|---|---|
| Reference | Positive Control | 0.543 | 0.522 | 0.402 | 0.775 | 0.939 | 0.000 | 0.000 |
| One-step | ProteoGAN | 0.376 | 0.093 | 0.121 | 0.510 | 0.277 | **0.095** | **0.055** |
| | ProGen2 | 0.414 | 0.355 | 0.240 | 0.663 | 0.545 | 0.109 | 0.064 |
| | CFP-Gen | 0.429 | 0.370 | 0.245 | 0.674 | 0.601 | 0.112 | 0.060 |
| Two-step | Chroma | 0.262 | 0.067 | 0.076 | 0.501 | 0.018 | 0.313 | 0.183 |
| | Pinal | 0.452 | 0.369 | 0.229 | 0.663 | 0.379 | 0.223 | 0.131 |
| Ours | CodeFP | **0.496** | **0.446** | **0.321** | **0.724** | **0.658** | 0.106 | 0.063 |

duce an effective inductive bias, facilitating the learning of functional semantics.

### 3.3. Local Structure-Function Supervision

While co-generative discrete diffusion models effectively capture the joint distribution of sequence and structure, optimizing them for functional constraints remains challenging due to the training ambiguity induced by the quantization discrepancy inherent in structural tokenizers. Unlike standard approaches that supervise solely on discrete outputs, we apply supervision directly to the CodeFP's continuous hidden states.

**Formulation.** During training, let $\mathbf{H}^{(L)} \in \mathbb{R}^{T \times d}$ be the continuous hidden states from the last transformer layer. For a protein annotated with GO label $y$, we first identify the indices of the structure tokens corresponding to the functional domain, using the same IPS-based localization described in Section 3.2. To facilitate downstream classification, we aggregate the hidden states at the relevant indices via mean pooling to obtain a continuous proxy for the functional domain's structure. We then employ a parameterized classifier head to project this embedding directly into the functional label space, yielding the logits corresponding to the $C$ GO terms. The classifier head and CodeFP are optimized jointly, a process facilitated by the frozen model decoder, which ensures that the learned embeddings remain aligned with the distribution of natural proteins.

**Class-Imbalanced Optimization.** To mitigate the impact of the long-tailed distribution in functional terms, we employ a mean-normalized inverse class frequency strategy. Specifically, we assign a scaling factor $w_c = N_c^{-1}/(\frac{1}{C}\sum_j N_j^{-1})$, where $N_i$ means the training set frequency for GO $i$, to balance the dominance of head classes. The final objective is minimized via weighted cross-entropy, defined as $\mathcal{L}_{\text{LSFS}} = -w_y \log \hat{p}_y$ for a target class $y$.

**Total Training Objective.** The derived auxiliary loss $\mathcal{L}_{\text{LSFS}}$ is integrated into the generative objective $\mathcal{L}_{\text{gen}}$ (defined in Equation 4). The total optimization objective is thus formu-

lated as:

$$\mathcal{L}_{\text{total}} = \mathcal{L}_{\text{gen}} + \gamma \mathcal{L}_{\text{LSFS}}, \quad (6)$$

where $\gamma$ serves as a balancing coefficient that scales the gradient contribution of the functional supervision. By imposing supervision on the latent space, CodeFP facilitates the incorporation of precise functional supervision signals, alleviating the training ambiguity.

## 4. Experiment

### 4.1. Experiment Setup

**Dataset.** We adopt the dataset collected by Yin et al. (2025), which comprises 103.9K protein sequences annotated with 375 GO terms derived from SwissProt (uni, 2025) and InterPro (Blum et al., 2025). The dataset is split into 95.6K training, 831 validation, and 8.3K testing samples. The test set contains 435 unique GO label combinations, including 76 combinations not observed during training. To facilitate structure-sequence co-generation, we first retrieve the precomputed structure tokens of $\sim 50$K sequences that overlap with DPLM-2's (Wang et al., 2024b) training set. Then, we query the PDB (Berman et al., 2000) and AlphaFoldDB (Varadi et al., 2024) databases to obtain the experimentally resolved or predicted 3D structures of remaining proteins and perform tokenization using DPLM-2's pre-trained LFQ encoder. We filter out 1.4K samples without a publicly available structure.

**Implementation Details.** We initialize CodeFP from the pre-trained DPLM-2 (650M) and apply FSR cross-attention to each layer of the Transformer block. We additionally inject one-hot GO embeddings via a gated network (Yin et al., 2025) and a cross-attention module. We train these added modules and the LSFS prediction head while keeping the remaining parameters frozen. During sampling, we draw the length of the protein uniformly between 200 and 400, and adopt the same procedure as DPLM-2 to obtain the structure tokens and amino acid tokens using 500 diffusion steps. The model is trained for approximately 60 epochs, taking about 48 hours, and achieves an inference latency of about

*Table 2.* **Foldability evaluation.** We report the structural success rates predicted by ESMFold.

| Model | pLDDT $> 70\,(\%)$ | pTM $> 0.5\,(\%)$ |
|---|---|---|
| Chroma | 23.47 | 66.76 |
| CFP-Gen | 75.52 | 72.30 |
| Pinal | 74.22 | 82.22 |
| CodeFP (Ours) | **80.65** | **83.48** |

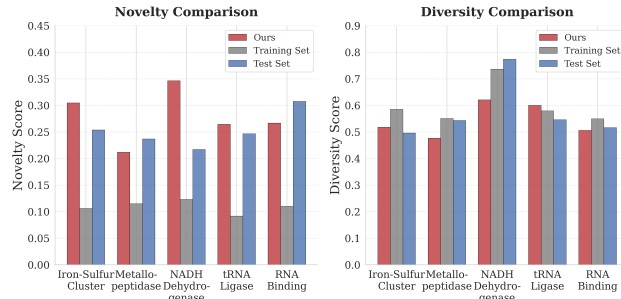

*Figure 3.* **Analysis of generative novelty and diversity.** We illustrate the distribution of Novelty (left) and Diversity (right) across five diverse functional tasks.

one minute per protein on a NVIDIA A800 GPU. Detailed hyperparameter settings are provided in Appendix D.

**Baselines.** We benchmark against five representative methods spanning two paradigms: (1) One-step generation, including ProteoGAN (Kucera et al., 2022) that adopts a conditional GAN as the backbone, ProGen2 (Madani et al., 2020) that leverages an autoregressive PLM to generate the sequence, and CFP-Gen (Yin et al., 2025) that performs discrete diffusion on amino acid sequences. (2) Two-step generation, including Chroma (Ingraham et al., 2023) that uses continuous diffusion and Pinal(Dai et al., 2024) that generate discrete structure tokens.

**Metrics.** We evaluate our model across three primary dimensions: functionality, foldability, and generative distribution, following (Yin et al., 2025). (1) *Functionality*: We assess functional fidelity from two perspectives. First, we employ Mean Reciprocal Rank (MRR) that directly evaluates the sequence similarity between generated proteins and ground-truth functional analogs. Second, we apply DeepGO-SE (Kulmanov et al., 2023) to predict GO terms based on generated sequences. We compare the predicted GO terms with the desired functions and report F1-Micro, F1-Macro, AUPR, and AUC-ROC scores. We also report Exact and Partial Match rates, which quantify whether all or any of the desired functions are recovered based on DeepGO's predictions. (2) *Foldability*: To assess whether sequences adopt stable, physically realizable conformations, we employ ESMFold (Lin et al., 2022) for structure prediction. A sequence is considered structurally successful if it achieves a mean pLDDT score above 70, indicating reliable local confidence, and a pTM score above 0.5, reflecting a consistent global fold. (3) *Generative Distribution*: We evaluate the generative distribution with diversity, novelty, and adherence to the natural protein distribution. Diversity captures variability among generated sequences and is defined as one minus the mean pairwise sequence identity. Novelty quantifies dissimilarity from the training set and is computed as one minus the maximum sequence identity to any training protein using MMseqs2 (Steinegger & Söding, 2017); higher values indicate better performance for both metrics. In addition, we assess distributional alignment with natural protein sequences using Maximum Mean Discrepancy (MMD) and its Gaussian-kernel variant (MMD-G), where

lower values denote closer alignment. Implementation details are provided in Appendix A.

### 4.2. Main Results

**CodeFP achieves superior functionality performance.** As shown in Table 1, our model surpasses Pinal in F1-micro (0.496 vs. 0.452) and outperforms CFP-Gen in both AUC-ROC (0.724 vs. 0.674) and MRR (0.658 vs. 0.601). These improvements indicate that the generated proteins more accurately capture functional specificity while reducing spurious assignments, and better align with functional analogs observed in natural proteins, highlighting the model's ability to capture intrinsic relationships between functional semantics and protein sequences.

**Improved coverage of long-tailed functional categories.** The performance gap widens on imbalance-sensitive metrics, with substantial gains over CFP-Gen in both F1-macro (0.446 vs. 0.370) and AUPR (0.321 vs. 0.245), indicating improved modeling of long-tailed functional categories. In contrast to baselines that tend to favor high-frequency functional modes, our method exhibits effective generalization to the long-tailed distribution, consistent with the complementary effects of FSR in facilitating functional abstraction and LSFS in reducing training ambiguity.

**State-of-the-art foldability of generated proteins.** As reported in Table 2, our model achieves the highest success rates, surpassing Pinal in both pLDDT (80.65% vs. 74.22%) and pTM (83.48% vs. 82.22%). These results indicate that the generated proteins are more likely to fold into well-defined structures, consistent with the benefits of jointly modeling sequence and structure.

**Preservation of natural protein distribution.** Consistently, all one-step generation models exhibit comparable MMD scores, whereas two-step approaches suffer substantially worse distributional alignment. Our model achieves competitive MMD (0.106 vs. 0.095) and MMD-G (0.063

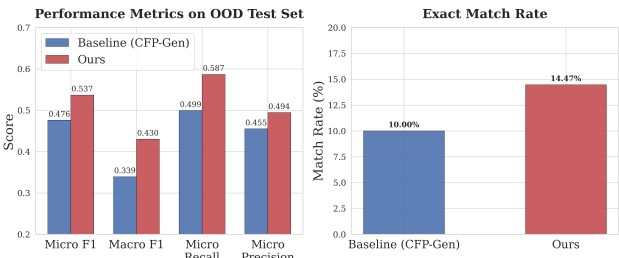

*Figure 4.* **Performance on OOD functional combinations.** We report multi-label classification metrics and the exact match rate on the OOD test subset.

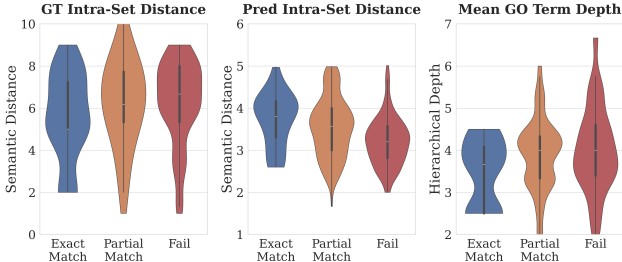

*Figure 5.* **Attribute analysis of generation difficulty.** We analyze generation outcomes against three topological attributes of the GO graph.

vs. 0.055) against ProteoGAN, indicating that improved functional controllability does not compromise alignment with the natural protein distribution. We attribute this performance to the sequence priors inherited from large-scale sequence pretraining.

**Novelty and diversity.** To assess the ability of our model to generate novel yet functional protein sequences under a constrained inference budget, we analyze five functional combinations spanning diverse biological mechanisms. For each combination, we generate 30 sequences and compare them with equal-sized samples drawn from the training and test sets. As shown in Fig. 3 (left), our model consistently achieves novelty scores substantially higher than the training set and generally comparable to those of the natural test set, indicating that CodeFP leverages retrieved results as functional priors to explore new regions of the sequence space. Meanwhile, Fig. 3 (right) shows that the diversity of the generated sequences closely matches the natural diversity observed in both training and test sets, indicating sufficient exploration of the sequence space.

### 4.3. Generalization to Out-of-Distribution Functional Combinations

We evaluate model generalization across two Out-of-Distribution (OOD) scenarios: (1) Unseen Natural Combinations, which occur in nature but are withheld from the training data, and (2) Hypothetical Combinations, which violate natural co-occurrence patterns and have no known biological instances.

**Unseen Natural Combinations.** We curate a test set of 76 functional combinations held out during training, generating 10 candidates per combination for assessment. As shown in Fig. 4, the low Exact Match Rates (peaking at only 14.5%) highlight the inherent difficulty of precisely realizing novel functional pairings. Nevertheless, the relatively high F1 and Recall scores indicate that the model captures partial functional constraints. Despite these challenges, our model outperforms the baseline by 9.1% in F1-Macro and 4.47% in Exact Match Rate, demonstrating superior zero-shot syn-

thesis capabilities.

To further elucidate the mechanisms underlying OOD generalization, we analyze model performance with respect to GO graph topology (Fig. 5). Specifically, semantic distance measures functional dissimilarity between GO terms, while term depth captures functional specificity, with deeper terms corresponding to more specialized functions. We observe that successful generations are associated with smaller semantic distances between target terms, indicating that functional similarity facilitates compositional synthesis. In contrast, failures are linked to greater GO term depth, suggesting that highly specific functions are more difficult to integrate. Moreover, these failures exhibit reduced predicted semantic diversity, reflecting a collapse toward a narrower and functionally homogeneous semantic space.

**Hypothetical Functional Combinations.** We further challenge the model to explore previously undefined regions of the functional landscape by generating 10 candidates per combination for 119 synthetically constructed hypothetical combinations (see Appendix C). Unfortunately, no generated protein fully satisfies the complete set of constraints, highlighting the inherent difficulty of engineering biologically viable proteins for artificial functional constraints. Nevertheless, our model exhibits partially correct functional generation even in this severe OOD setting (Fig.6). It significantly outperforms the baseline, raising the F1 score to 0.330 (vs. 0.174) and the Partial Match Rate to 43.20% (vs. 5.54%).

### 4.4. Case Study

To provide an intuitive illustration of the model's generative capability, we present a representative case study on protein generation conditioned on an OOD functional combination from the test set. As illustrated in Fig. 7, our model successfully generates well-formed local structural motifs that closely resemble the functional motifs observed in natural proteins. In contrast, the baseline fails to fold into a structured protein, with this extreme case collapsing entirely into disordered coils lacking defined secondary structure.

*Table 3.* **Ablation study on component contributions.** We analyze the impact of co-generation, FSR, and LSFS on functional consistency, distributional alignment, and foldability.

| Model Variant | Functional Consistency & Distribution | | | | Structural Realizability | |
|---|---|---|---|---|---|---|
| | F1-Micro (↑) | F1-Macro (↑) | MRR (↑) | MMD (↓) | pLDDT > 70 (%) | pTM > 0.5 (%) |
| CodeFP | **0.496** | **0.446** | 0.658 | 0.106 | 80.65 | **83.48** |
| w/o LSFS | 0.495 | 0.437 | 0.645 | 0.172 | **82.01** | 81.73 |
| w/o FSR | 0.486 | 0.423 | **0.674** | **0.101** | 71.57 | 76.76 |
| w/o FSR & LSFS | 0.465 | 0.400 | 0.534 | 0.192 | 71.71 | 71.98 |
| two-step generation | 0.414 | 0.285 | 0.312 | 0.282 | 52.24 | 59.54 |

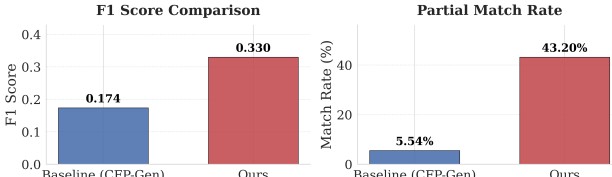

*Figure 6.* **Performance on Hypothetical Functional Combinations.** We evaluate the ability to generate proteins for 119 functional combinations not found in nature.

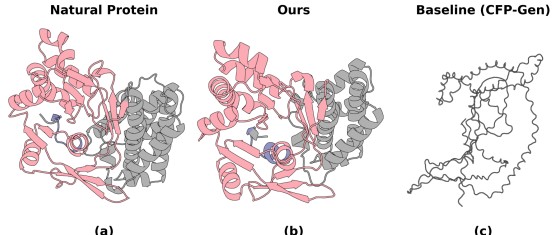

*Figure 7.* **Visualization of Multi-Functional Protein Generation for OOD Combinations.** We generate a protein conditioned on the unseen functional combination of *Mannitol-1-phosphate 5-dehydrogenase activity* (GO:0008926) and *NAD binding* (GO:0051287). (a) Natural protein structure. (b) Structure generated by our method. (c) Structure generated by the baseline CFP-Gen. The *NAD-binding* structural motif is highlighted in red, while the catalytic motif associated with *Mannitol-1-phosphate 5-dehydrogenase activity* is shown in blue.

Complementing this visual analysis, quantitative metrics further confirm the quality of our generation. The generated protein achieves a pLDDT of 94.9 and a pTM of 0.96, indicating excellent foldability. Crucially, the maximum sequence identity compared to proteins of the same function is merely 32%. This low homology denotes significant sequence novelty and underscores the model's capacity for *de novo* functional integration.

### 4.5. Ablation Study

To dissect the contributions of our individual components, we evaluate a two-step version and variants excluding the Functional-Structural Retrieval (FSR) and Local Structure-Function Supervision (LSFS) modules (Table 3).

**Efficacy of Co-generation.** The ablation results demonstrate that CodeFP (w/o FSR, LSFS) consistently surpasses the one-step baseline (CFP-Gen) in functionality while exhibiting superior foldability than the two-step version. This validates our hypothesis: explicitly modeling the joint probability offers a superior foundation for functional design compared to one-step generation, while preserving foldability more readily than two-step generation.

**Dual Role of FSR.** The integration of FSR yields simultaneous gains in functional metrics (F1-Micro: 0.495 vs. 0.465) and foldability (pLDDT > 70: 82.01% vs. 71.71%). This dual gain suggests that retrieved structural motifs serve as an essential inductive bias, effectively ground functional semantics, and facilitate the holistic functional co-generation of protein sequence and structure.

**Distributional Alignment via LSFS.** The deployment of LSFS substantially improves MRR (0.674 vs. 0.534) and reduces MMD (0.101 vs. 0.192). These distributional shifts confirm that the generative distribution aligns more closely with the natural functional protein space, indicating that LSFS provides precise functional supervision that effectively captures the functional semantics inherent in natural proteins.

Our Full Model effectively integrates these mechanisms, yielding the highest functional performance without compromising foldability or distributional fidelity.

### 5. Conclusion and Future Work

In this work, we introduce **CodeFP**, a novel co-generative PLM framework that unifies sequence and structure generation to advance *de novo* functional protein design. To extend co-generation to functional design, we propose two critical mechanisms: Functional-Structural Retrieval (FSR), which grounds function semantics by structure motifs, and Local Structure-Function Supervision (LSFS), which mitigates training ambiguity via latent space supervision. Empirical evaluations on benchmarks demonstrate that CodeFP achieves state-of-the-art performance in both functional con-

sistency and structural foldability, with ablation studies validating the necessity of each component.

While CodeFP advances *de novo* functional protein design, conditioning generation on OOD functional combinations remains challenging. We expect future works on (1) augmenting the current dataset to encompass a broader spectrum of functions, thereby facilitating more rigorous evaluation benchmarks (2) investigating novel function combination design to enhance robustness against OOD shifts (3) applying CodeFP to wet-lab validation empirically substantiate its practical utility and reliability.

## Code Availability

CodeFP is available in the OpenBioMed toolkit: `https://github.com/PharMolix/OpenBioMed`.

## Acknowledgements

This research is supported by the Innovative Drug Research and Development National Science and Technology Major Project (No. 2025ZD1803101) and PharMolix Inc.

## Impact Statement

This paper presents work whose goal is to advance the field of Machine Learning. There are many potential societal consequences of our work, none which we feel must be specifically highlighted here.

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

# A. Evaluation Metrics

We comprehensively evaluate the generated proteins across three dimensions: Sequence Plausibility (distributional similarity to natural proteins), Functional Consistency (alignment with target functional constraints), and Structural Realizability (physical foldability).

## A.1. Sequence Distribution Metrics

To quantify how well the generated proteins capture the biophysical properties of natural proteins, we measure the distributional discrepancy between the full set of generated sequences $\mathcal{G}$ and natural sequences $\mathcal{P}$. We utilize Maximum Mean Discrepancy (MMD) with sequence embeddings derived from normalized Spectrum Mapping (k-mer frequencies). We report MMD with two kernels:

- **Linear MMD ($\text{MMD}_{\text{lin}}$):** Measures the Euclidean distance between the mean embeddings of the real and generated distributions:

$$\text{MMD}_{\text{lin}}(\mathcal{P}, \mathcal{G}) = |\mu_{\mathcal{P}} - \mu_{\mathcal{G}}|2 \tag{7}$$

  where $\mu_{\mathcal{P}}$ and $\mu_{\mathcal{G}}$ are the means of the sequence embeddings.

- **Gaussian MMD ($\text{MMD}_{\text{rbf}}$):** Incorporates a Radial Basis Function (RBF) kernel $k(x, y) = \exp(-\gamma\|x - y\|^2)$ to capture higher-order distributional moments. The bandwidth $\gamma$ is determined via the median heuristic.

Lower MMD values indicate that the generated sequences share similar statistical properties with natural proteins.

## A.2. Functional Consistency Metrics

To assess whether the generated sequences satisfy the specified functional conditions, we employ two complementary evaluation strategies: oracle-based classification and distribution-based ranking.

**Oracle-based Metrics.** We utilize a pre-trained state-of-the-art function prediction model as an oracle to classify the generated sequences. By comparing the predicted labels against the input conditional labels, we report the following standard metrics:

- **Macro/Micro F1-score:** To balance performance across classes with varying frequencies, we report both Macro-F1 (arithmetic mean of per-class F1) and Micro-F1 (global calculation based on total true/false positives/negatives).

- **Macro AUPR & AUC:** We compute the Area Under the Precision-Recall Curve (AUPR) and the Receiver Operating Characteristic Curve (AUC), averaged across all classes.

**Distribution-based Metric.** We evaluate the distributional alignment between generated and natural sequences within the same functional category. We utilize a **Mean Reciprocal Rank (MRR)** metric based on the Maximum Mean Discrepancy (MMD). Let $\mathcal{P}_c$ and $\mathcal{G}_c$ denote the sets of real and generated sequences, respectively, for a specific function label $c \in \{1, \ldots, C\}$. We compute the linear MMD distance between the real set of class $c$ ($\mathcal{P}_c$) and the generated sets of all classes ($\mathcal{G}_{c'}$ for all $c'$). If the generation is distinct and accurate, $\mathcal{G}_c$ should be closest to $\mathcal{P}_c$. The MRR is defined as:

$$\text{MRR}(\mathcal{G}, \mathcal{P}) = \frac{1}{C} \sum_{c=1}^{C} \frac{1}{\text{rank}_{\mathcal{G}}(\text{MMD}(\mathcal{G}c, \mathcal{P}c))} \tag{8}$$

where $\text{rank}_{\mathcal{G}}(\cdot)$ is the rank of the distance $\text{MMD}(\mathcal{G}_c, \mathcal{P}_c)$ among the set of distances $\{\text{MMD}(\mathcal{G}_{c'}, \mathcal{P}_c)\}_{c'=1}^{C}$. An MRR of 1.0 indicates perfect functional mode matching.

## A.3. Structural Realizability Metrics

Since the generated outputs are primary sequences, we assess their foldability by predicting their 3D structures using ESMFold. We utilize two confidence metrics provided by the folding engine:

- **pLDDT:** The predicted Local Distance Difference Test score. We calculate the mean pLDDT per protein. A score $> 70$ indicates a high-confidence prediction, suggesting the sequence adopts a stable local structure.

- **pTM:** The predicted Template Modeling score, which estimates the global topological accuracy. We consider sequences with pTM $> 0.5$ as having a likely correct global fold.

### A.4. Robustness to Alternative Evaluation Oracles

To confirm that our findings are not artifacts of the specific predictive models used in the main text, we assess the generated sequences using alternative evaluation oracles. We benchmark CodeFP against the leading baselines from both the one-step (CFP-Gen) and two-step (Pinal) paradigms.

**Functional Evaluation via NetGO 4.0.** We re-evaluated the full test set (8,309 sequences) using NetGO 4.0 (Yan et al., 2025). To ensure prediction reliability, we filtered out predicted GO terms with confidence scores below 0.6, aligning with the NetGO 4.0 guidelines for high-confidence predictions. As shown in Table 4, CodeFP's performance advantage becomes even more pronounced under NetGO 4.0, yielding a 0.151 absolute improvement in F1-Micro over CFP-Gen.

*Table 4.* Functional consistency evaluated by NetGO 4.0.

| Model | F1-Micro (↑) | F1-Macro (↑) | AUPR (↑) | AUC-ROC (↑) |
|---|---|---|---|---|
| Pinal | 0.479 | 0.486 | 0.348 | 0.766 |
| CFP-Gen | 0.511 | 0.538 | 0.421 | 0.810 |
| CodeFP (Ours) | **0.662** | **0.683** | **0.591** | **0.868** |

**Structural Evaluation via OmegaFold.** Due to the high computational cost of OmegaFold (Wu et al., 2022) inference, we randomly sampled 1,000 sequences from the test set for structural prediction. As reported in Table 5, CodeFP maintains its advantage in structural foldability. The consistent trends observed across independent predictive oracles verify the robustness of our evaluation metrics and the empirical significance of the proposed method.

*Table 5.* Structural foldability evaluated by OmegaFold on a 1,000-sample subset.

| Model | pLDDT $> 70$ (↑) |
|---|---|
| CFP-Gen | 70.10% |
| Pinal | 75.70% |
| CodeFP (Ours) | **78.50%** |

## B. Extensive Analysis

To gain deeper insights into the generative behavior of CodeFP and further characterize its performance boundaries, we conduct a multifaceted analysis based on the semantic properties of the functional labels, the distributional properties of the training data, and the empirical outcomes of the generated sequences. Specifically, our analysis spans four key dimensions: (i) the influence of functional semantic difficulty on model performance; (ii) the impact of biological and distributional properties on performance gaps; (iii) the investigation of pretraining data overlap and model generalization; and (iv) a detailed characterization of failure modes regarding foldability versus functionality.

### B.1. Performance vs. Semantic Difficulty and Oracle Bias

We first analyze how model performance (F1-score and Recall) varies with the Semantic Difficulty of the input condition. We define the input's semantic difficulty as the mean semantic distance of the requested GO label combination. The semantic distance between two GO labels is calculated as the shortest path length on the GO Directed Acyclic Graph (DAG), where edges represent "is-a" relationships. For a set of input labels, the mean distance is the average of pairwise distances between all labels in the set.

As shown in Fig.8, the fluctuation in F1-score and Recall for both our model and the baseline (CFP-Gen) does not strictly correlate with increased difficulty. Instead, it exhibits a strong correlation with the performance of the Reference (Oracle) model. Notably, in the region where the mean semantic distance is 4–5, both the baseline and our model suffer a significant performance drop. This decline coincides with a sharp drop in the Reference model's performance. This suggests that the

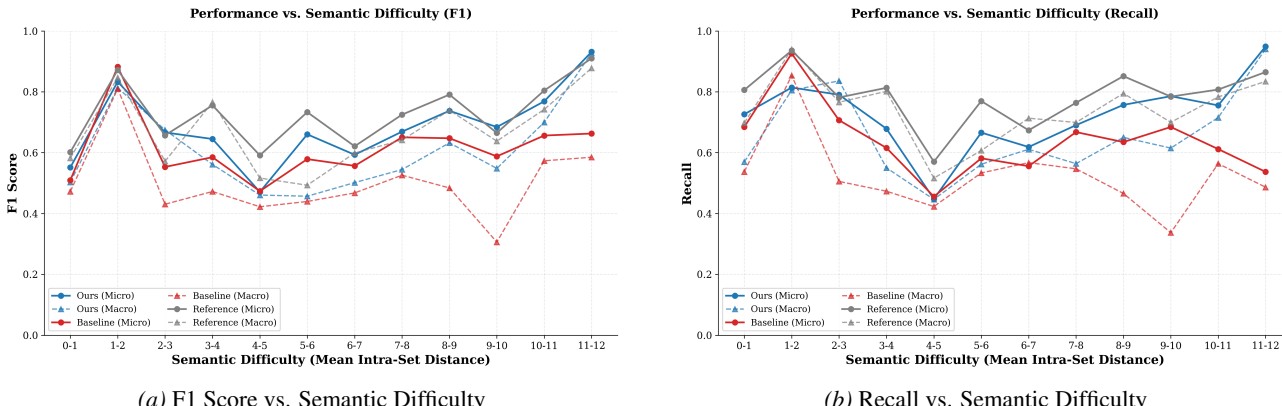

(a) F1 Score vs. Semantic Difficulty

(b) Recall vs. Semantic Difficulty

*Figure 8.* **Performance variation across Semantic Difficulty.** The x-axis represents the mean intra-set semantic distance of input GO labels. We observe a strong correlation between the generative models' performance and the Reference (Oracle) model's performance.

current evaluation metrics are potentially bottlenecked by the capability of the functional predictor (Oracle), limiting the assessed performance of generative models in specific semantic regions.

### B.2. Analysis of Performance Gap Across Distributional and Biological Properties

Given the potential bias in absolute evaluation metrics identified above, we further analyze the **Performance Gap ($\Delta$)**, defined as the score difference between the generative model and the Reference. We compare our model against the baseline across five metrics spanning data distribution and biological significance:

1. **Co-occurrence Strength (Typicality):** Measures how often label pairs appear together. For a pair of labels, it is calculated as their intersection count in the training set divided by the sum of their individual counts.

2. **Train Frequency:** The $\log_{10}$ of the total occurrence count of the label in the training set.

3. **Specificity (IDF):** The Inverse Document Frequency, treating GO labels as words and proteins as documents, calculated as $\log_{10}(N/\text{count})$, where $N$ is the total number of training samples.

4. **Semantic Difficulty (Avg Distance):** As defined in B.1.

5. **Annotation Specificity (Avg Depth):** The average depth of the labels in the GO hierarchy, defined as the shortest path distance from the root node (GO:0003674).

As illustrated in Fig.9, our model consistently exhibits higher $\Delta$ values than the baseline across the majority of metric intervals. The performance trends of our model generally mirror those of the baseline, particularly in Co-occurrence Strength and Depth.

### B.3. Impact of Pretraining Data Overlap

Among the 8,309 test sequences, 3,907 sequences were present in the DPLM-2 pretraining corpus. However, this does not constitute data leakage for our specific task. DPLM-2 pretraining optimizes for the unconditional distribution $p(\mathcal{P})$. In contrast, *de novo* functional protein design evaluates the conditional generative capability $p(\mathcal{P}|c_{GO})$. Exposure to a sequence during unconditional pretraining does not equip the model to learn the complex mapping between functions and their specific sequence-structure instantiations.

To demonstrate that our model's performance does not rely on memorized unconditional distributions, we filtered out the 3,907 overlapping sequences and re-evaluated the models on the remaining 4,402 strictly unseen test proteins. As shown in Table 6, CodeFP maintains its substantial lead in both functional consistency (e.g., F1-Micro +0.042 over baselines) and structural realizability (e.g., pLDDT > 70 +5.00% over CFP-Gen).

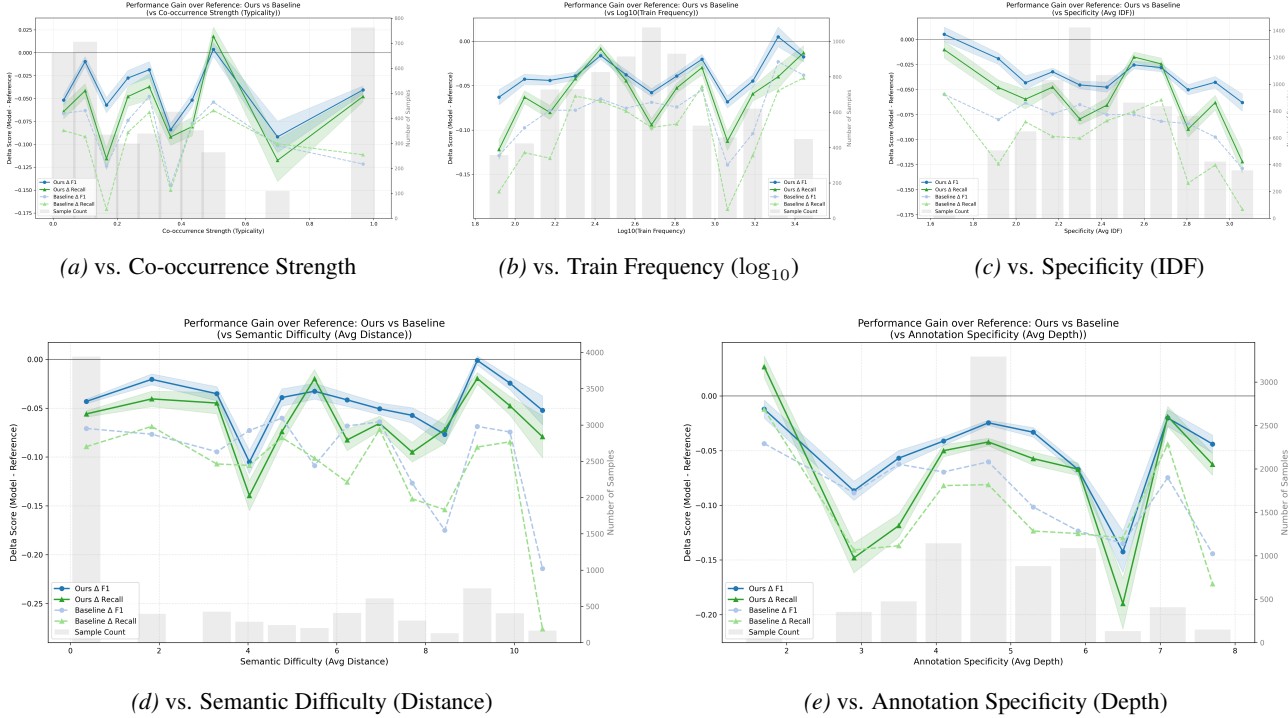

*(a)* vs. Co-occurrence Strength

*(b)* vs. Train Frequency ($\log_{10}$)

*(c)* vs. Specificity (IDF)

*(d)* vs. Semantic Difficulty (Distance)

*(e)* vs. Annotation Specificity (Depth)

*Figure 9.* **Delta Performance ($\Delta$F1 and $\Delta$Recall) across different functional properties.** The curves represent the performance gap relative to the Reference. Our model (solid lines) consistently achieves a smaller gap (higher values) compared to the baseline (dashed lines).

*Table 6.* Performance on the strictly unseen test subset (4,402 sequences).

| Model | F1-Micro ($\uparrow$) | F1-Macro ($\uparrow$) | AUPR ($\uparrow$) | AUC-ROC ($\uparrow$) | pLDDT $> 70$ ($\uparrow$) | pTM $> 0.5$ ($\uparrow$) |
|---|---|---|---|---|---|---|
| Pinal | 0.448 | 0.377 | 0.237 | 0.668 | 71.51% | **79.64%** |
| CFP-Gen | 0.448 | 0.409 | 0.285 | 0.697 | 73.28% | 68.94% |
| CodeFP (Ours) | **0.490** | **0.452** | **0.328** | **0.727** | **78.28%** | 79.05% |

### B.4. Failure Mode Analysis: Foldability vs. Functionality

To understand the primary bottlenecks when the model fails to generate valid targets, we analyzed the relationship between structural and functional success. We plotted the joint density distribution of foldability (measured by pLDDT) versus functionality (measured by F1-Score) for every individual sample generated in the test set.

As illustrated in Figure 10, the empirical distribution clearly shows a concentration of high pLDDT scores across varying F1-Scores. This indicates that even when the model struggles to perfectly satisfy complex functional constraints, it successfully maintains stable structural geometries. Consequently, CodeFP tends to fail on functionality rather than foldability. This highlights that translating abstract functional conditions into exact sequence-structure mappings remains the primary challenge in *de novo* functional protein design.

## C. Construction of the Hypothetical Function Combination Dataset

**Candidate Selection via Structural Conservation.** We curate a pool of 21 GO terms exhibiting high motif structural stability, specifically selecting those with a median Root Mean Square Deviation (RMSD) below 0.5 Å in the training set. This criterion ensures that the selected functions correspond to highly conserved local structures. From this pool, we generate pairwise combinations through random sampling, filtering out any pairs that co-occur in the training distribution. This procedure yields a final test set of 119 hypothetical GO label combinations.

**Dataset Statistics.** The resulting dataset exhibits a mean semantic distance of 8.7 and a mean semantic depth of 5.1. These

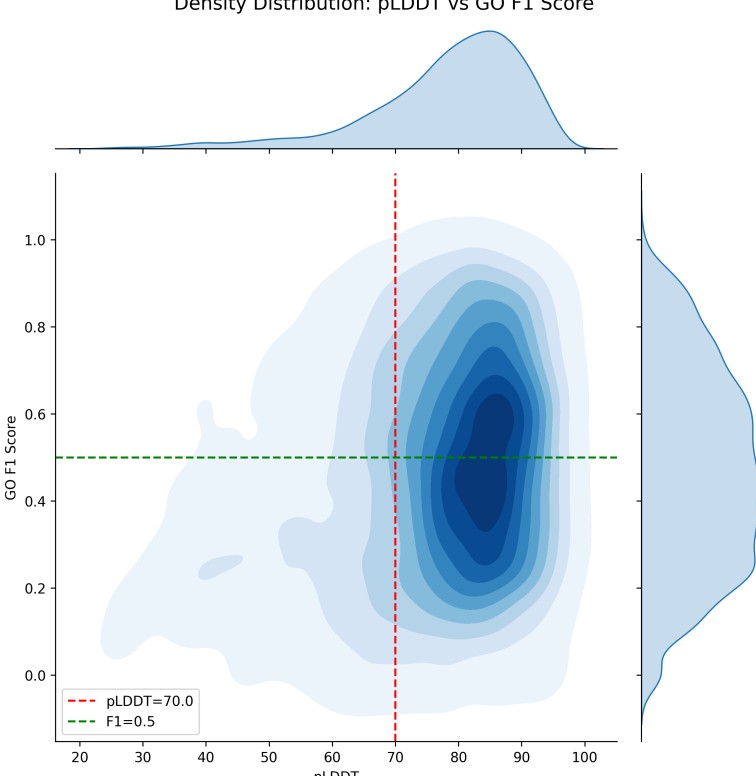

*Figure 10.* **Joint density distribution of foldability (pLDDT) and functionality (F1-Score).** The empirical distribution demonstrates that CodeFP robustly generates foldable structures even when functional consistency is suboptimal.

statistics indicate that the selected combinations possess high functional specificity (high depth) while maintaining weak functional correlation (large semantic distance).

## D. Model Details, Training Costs, and Hyperparameters

**Model Architecture and Parameter Breakdown.** CodeFP scales to a total of approximately 1.87B parameters. The architectural breakdown consists of: the frozen DPLM-2 backbone ($\sim$670M), a gated network and cross-attention module used to inject one-hot GO embeddings following CFP-Gen ($\sim$930M), and the Functional-Structural Retrieval (FSR) module ($\sim$270M).

**Training Setup and Computational Costs.** The training process was executed on a cluster of 4 NVIDIA A800 GPUs for approximately 60 hours. Optimization was performed using AdamW with a peak learning rate of $4 \times 10^{-5}$ and a global batch size of 2,048 tokens; all other hyperparameters remain consistent with the base model configuration. Additionally, constructing the FSR database incurs a one-time offline computational cost. Sequence processing via InterProScan averages $\sim$15.6 seconds per protein on an Intel Xeon 8374C CPU, which is trivially parallelizable. Subsequent GPU operations (LFQ encoding and [CLS] extraction) achieve high throughputs of $\sim$5,800 and $\sim$3,900 residues per second on a single A800 GPU requiring under 10GB of VRAM. Finally, the GO-protein aggregation involves lightweight CPU vector operations with negligible overhead.

**Impact of LSFS Configuration.** We investigate the sensitivity of Local Structure-Function Supervision (LSFS) to the weighting coefficient $\gamma$ and class-frequency weighting (Table 7). A key finding is that unweighted LSFS suffers from poor tail-class performance; applying frequency-based weighting ($\gamma = 1.0$) substantially improves F1-Macro from 0.374 to 0.419, effectively mitigating optimization bias in the long-tailed functional distribution. Furthermore, increasing $\gamma$ to 2.0 yields consistent gains in both metrics, suggesting that LSFS offers an accurate supervision signal that guides functional alignment.

*Table 7.* **Impact of LSFS Hyperparameters.** We evaluate the sensitivity of functional alignment (F1-Micro and F1-Macro) to the LSFS loss weighting coefficient ($\gamma$) and the class-frequency weighting strategy.

| CONFIGURATION | $\gamma$ (LSFS) | F1-MICRO | F1-MACRO |
|---|---|---|---|
| LSFS (UNWEIGHTED) | 1.0 | 0.465 | 0.374 |
| LSFS (WEIGHTED) | 1.0 | 0.474 | 0.419 |
| LSFS (WEIGHTED) | 2.0 | 0.486 | 0.423 |

**Exploration of Backbone Fine-Tuning.** We explored the impact of fine-tuning the DPLM-2 backbone using Low-Rank Adaptation (LoRA) for 30K steps. As shown in Table 8, evaluation on a randomly sampled 1K test subset indicated a slight performance degradation compared to keeping the backbone frozen. This drop suggests that the model might be adapting pre-trained representations in a way that disrupts natural sequence priors, rather than effectively learning new functional mappings.

*Table 8.* Performance comparison of frozen versus LoRA-finetuned backbone on a 1K test subset.

| Setup | F1-Micro ($\uparrow$) | F1-Macro ($\uparrow$) | AUPR ($\uparrow$) | AUC-ROC ($\uparrow$) |
|---|---|---|---|---|
| CodeFP (Frozen Backbone) | **0.490** | **0.484** | **0.365** | **0.751** |
| CodeFP (LoRA Backbone) | 0.483 | 0.476 | 0.354 | 0.745 |

**Inference Guidance Mechanisms.** We evaluated the compatibility of CodeFP with inference guidance techniques. Standard Classifier Guidance (CG) is not feasible during the diffusion process, as the LSFS head requires fully formed local structures that cannot be accurately isolated mid-generation. Conversely, CodeFP natively supports Classifier-Free Guidance (CFG), having been trained with a 10% condition dropout rate. As detailed in Table 9, applying a CFG scale of 1.5 enforces stronger functional alignment (yielding improvements in F1 and pTM) with only a minimal trade-off in local structural confidence (pLDDT).

*Table 9.* Impact of Classifier-Free Guidance (CFG) on an 800-protein test subset.

| Inference Mode | F1-Micro ($\uparrow$) | F1-Macro ($\uparrow$) | pLDDT $> 70$ ($\uparrow$) | pTM $> 0.5$ ($\uparrow$) |
|---|---|---|---|---|
| Standard (Scale = 1.0) | 0.497 | 0.492 | **83.75%** | 84.50% |
| CFG (Scale = 1.5) | **0.507** | **0.501** | 81.12% | **86.12%** |

**Validity of the [CLS] Representation for FSR.** To represent structural semantics within the FSR module, we utilize the [CLS] token embeddings. While average pooling of structure tokens remains a viable alternative for future exploration, empirical analysis confirms that [CLS] embeddings effectively capture structural semantics. When grouping the [CLS] embeddings by their corresponding GO terms, we observed highly distinct clustering. This distinction is quantitatively supported by significant statistical test results, including PERMANOVA ($p = 0.001, F = 127.53$), ANOSIM ($p = 0.001, R = 0.704$), and $\eta^2 = 0.573$. These metrics demonstrate that the [CLS] token successfully encodes the essential geometric features of functional motifs.

**Inference Settings.** Inference utilizes a 500-step iterative sampling procedure. To dynamically modulate generation diversity, a linear temperature annealing schedule is applied, where the temperature $T_t$ decays from $T_{\max} = 2.0$ to $T_{\min} = 1.0$ according to $T_t = T_{\min} + (T_{\max} - T_{\min}) \cdot (1 - \frac{t}{N})$. Target sequence lengths are sampled uniformly from $U(200, 400)$.

# E. Implementation of Baselines

For ProteoGAN, ProGen2, and CFP-Gen, we adopt the experimental results directly from the CFP-Gen publication (Yin et al., 2025), as our study utilizes the identical training and evaluation data splits. This ensures a fair and direct comparison with established benchmarks.

For Chroma and Pinal, which support natural language conditioning, we facilitate comparison by transforming the structured GO label combinations in the test set into natural language descriptions. Specifically, we construct input prompts by embedding the function name and target length into a standardized template. A representative prompt used for inference is:

"*Generate a protein that functions as: 3-dehydroquinate dehydratase activity. The sequence length is approximately 285.*"

For Ours(two-step) in the ablation study, we first train exclusively on structure tokens. During inference, the predicted structure tokens are passed to a pre-trained DPLM-2 inverse folding model to generate the corresponding sequence tokens.

**ESM3 Baseline.** We evaluated the open-weights version of ESM3. Because ESM3 requires explicit sequence positions for InterPro (IPR) tracks, we mapped the target GO terms to their corresponding IPR identifiers and provided the ground-truth motif positions to condition the generation. As shown in Table 10, CodeFP significantly outperforms ESM3 across all functional metrics.

*Table 10.* Performance comparison between ESM3 and CodeFP on functional consistency.

| Model | F1-Micro (↑) | F1-Macro (↑) | AUPR (↑) | AUC-ROC (↑) |
|-------|--------------|--------------|----------|-------------|
| ESM3 | 0.221 | 0.044 | 0.040 | 0.508 |
| CodeFP | **0.496** | **0.446** | **0.321** | **0.724** |

**DPLM-2 Motif-Scaffolding Baseline.** To isolate the benefits of our architectural additions from the base model's inherent scaffolding capabilities, we evaluated a DPLM-2 Motif-Scaffolding baseline (i.e., without any architectural changes). For a random subset of approximately 400 test proteins, we extracted the ground-truth motif structure tokens corresponding to each GO term. To manage overly long motifs, the extracted motif length was capped at a maximum of 30 tokens per term. The remaining amino acid and structure tokens were masked and subsequently generated. As reported in Table 11, even when the baseline is provided with ground-truth structural motifs, CodeFP achieves superior functional consistency, validating the effectiveness of our proposed co-generative approach.

*Table 11.* Performance comparison on a ∼400 protein subset evaluating base scaffolding capabilities.

| Model | F1-Micro (↑) | F1-Macro (↑) | AUPR (↑) | AUC-ROC (↑) |
|-------|--------------|--------------|----------|-------------|
| DPLM-2 (Motif-Scaffolding) | 0.449 | 0.453 | 0.339 | 0.758 |
| CodeFP | **0.520** | **0.550** | **0.442** | **0.794** |

## F. Detailed Case Studies and Semantic Explanations

In this section, we provide detailed semantic definitions for the Gene Ontology (GO) term combinations utilized in our experiments and offer an in-depth qualitative analysis of representative generation outcomes, encompassing both successes and structural/functional failures.

### F.1. Target Groups for Novelty and Diversity Testing

To evaluate the model's capability to generate diverse structures within specific functional niches, we selected five distinct functional groups. As detailed in Table 12, these groups encompass a broad spectrum of biochemical activities, ranging from metal-sulfur cluster binding and electron transport to nucleic acid processing and enzymatic ligation.

### F.2. Analysis of Successful Generations

We present two representative case studies to illustrate how CodeFP resolves complex structural constraints and maintains robustness, even under out-of-distribution conditions.

**Case 1: OOD Generalization and Structural Robustness.** The example in Fig. 7 evaluates a functional combination drawn from the Out-of-Distribution (OOD) subset: mannitol-1-phosphate 5-dehydrogenase activity (`GO:0008926`) and NAD binding (`GO:0051287`). Because this pairing never co-occurs in the training corpus, it rigorously tests the model's compositional generalization. The specification imposes dual structural constraints: forming a catalytically competent active site for NAD(H)-dependent interconversion, while simultaneously constructing a defined binding pocket for the NAD cofactor.

*Table 12.* **Functional Target Groups.** Detailed breakdown of the GO term combinations used in the novelty and diversity experiments.

| Group ID | GO Terms | Biological Semantics |
|---|---|---|
| **Iron-Sulfur Cluster** | GO:0004076, GO:0005506, GO:0051537, GO:0051539 | Involves the binding of iron ions and 2Fe-2S clusters, playing critical roles in electron transfer and catalytic processes. |
| **Metallo-peptidase** | GO:0004477, GO:0004488 | Represents bifunctional enzymatic activities (methenyltetrahydrofolate cyclohydrolase and dehydrogenase) essential for the folate cycle and one-carbon metabolism. |
| **NADH Dehydrogenase** | GO:0008137, GO:0048038, GO:0050136 | Encompasses NADH dehydrogenase (quinone/ubiquinone) activity, central to the mitochondrial electron transport chain and cellular respiration. |
| **tRNA Ligase** | GO:0004070, GO:0016597 | Includes phosphopantothenoylcysteine decarboxylase and aminoacyl-tRNA ligase activities, fundamental for protein biosynthesis and coenzyme A metabolism. |
| **RNA Binding** | GO:0003723, GO:0004523, GO:0030145 | Covers broad RNA binding capabilities and specific ribonuclease activities (e.g., RNA-DNA hybrid digestion), regulating gene expression and RNA stability. |

While CodeFP successfully realizes these constraints, the baseline (CFP-Gen) exhibits a severe structural collapse (Fig. 7c). To quantitatively verify that this collapse is indicative of a broader empirical trend rather than an isolated outlier, we analyzed the lower-bound foldability across all generated sequences within the OOD subset. As shown in Table 13, CFP-Gen suffers from significant structural degradation, with 14.0% of its OOD generations yielding a pTM below 0.2. In contrast, CodeFP reduces this critical failure rate to merely 1.86%, underscoring the superior structural robustness of our co-generative approach when confronted with unseen functional combinations.

*Table 13.* Rates of severe structural degradation within the OOD test subset.

| Model | pTM $< 0.2$ ($\downarrow$) | pTM $< 0.3$ ($\downarrow$) |
|---|---|---|
| CFP-Gen (Baseline) | 14.0% | 22.4% |
| CodeFP (Ours) | **1.86%** | **9.57%** |

**Case 2: Bifunctional Motif Integration.** Fig. 11 visualizes the generation of a bifunctional enzyme involved in folate metabolism, constrained by GO:0004477 (methenyltetrahydrofolate cyclohydrolase activity) and GO:0004488 (methylenetetrahydrofolate dehydrogenase activity). This combination requires the model to generate a sequence capable of catalyzing two sequential biochemical reactions, entailing a structurally coordinated active site.

Structurally, CodeFP successfully integrates the distinct functional motifs required for both activities into a globally coherent and stable backbone, achieving a high confidence score (pLDDT: 94.37). Sequentially, the generated protein exhibits only 52.94% sequence identity to the natural reference sequence. This moderate homology confirms that CodeFP actively constructs distinct, viable *de novo* variants that preserve essential functional geometries, rather than simply memorizing evolutionary templates.

### F.3. Qualitative Analysis of OOD Failure Modes

To complement the quantitative hypotheses presented in Section 4.3—specifically that generation difficulty is tied to GO term depth and semantic distance—we examine two representative failure cases from the OOD subset. Both generated sequences achieved high structural confidence but failed to satisfy the functional constraints, highlighting the current boundaries of *de novo* functional design.

**Failure Case 1: Insufficient Specificity (High GO Depth).**

- **Target Functions:** dTTP/UTP pyrophosphatase activity (GO:0036221, GO:0036218, GO:0047429).

# Case Study: Conditional Generation for Target Q48KZ8

**Ours**

**Baseline (CFPGen)**

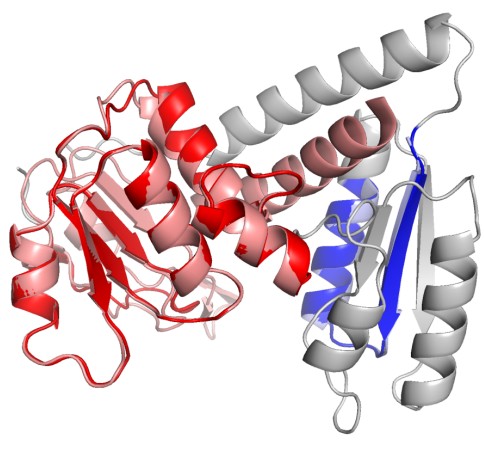
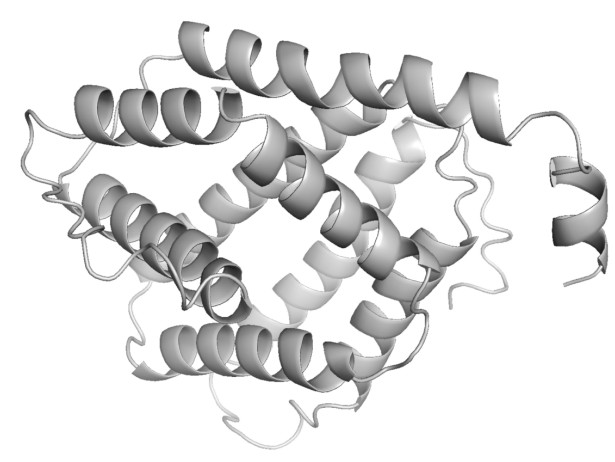

pLDDT: 94.37 | pTM: 0.941

pLDDT: 36.94 | pTM: 0.181

Conditioning GO Terms: GO:0004488, GO:0004477

*Figure 11.* **Structural generation for Target Q48KZ8.** We compare the structures generated by our model (left) and the baseline (right) conditioned on dual functional constraints (GO:0004477, GO:0004488). The specific local motifs required for both functions are highlighted in red and blue, respectively.

- **Predicted Functions:** Pyrophosphatase activity (GO:0016462), nucleotide binding (GO:0000166), ATP binding (GO:0005524), alongside other generic hydrolase terms.

- **Generated pLDDT:** 94.47

*Analysis:* The model correctly identifies the overarching catalytic mechanism (pyrophosphatase activity) and general nucleotide binding capability. However, it struggles with precise substrate specificity, predicting ATP binding instead of the targeted dTTP/UTP binding. This case illustrates that predicting deep, specific GO terms remains highly challenging. Such specificity likely relies on fine-grained, sub-angstrom structural nuances in the binding pocket that are difficult to accurately model without explicit atomic-level refinement.

**Failure Case 2: Semantic Disconnect (Large Semantic Distance).**

- **Target Functions:** Nicotinate-nucleotide-dimethylbenzimidazole phosphoribosyltransferase activity (GO:0008939) and nucleotide binding (GO:0000166).

- **Predicted Functions:** Catalytic activity (GO:0003824), general binding (GO:0005488), and broad transferase activity (GO:0016740).

- **Generated pLDDT:** 93.22

*Analysis:* While producing a highly stable backbone, the model defaults to generating generic, upper-level GO terms. It fails to physically reconcile the distinct and distant semantic combination of specific binding and transferase activities (semantic distance = 8). This validates our quantitative finding that larger semantic distances exacerbate compositional difficulty, causing the model to retreat to a more generalized, unspecialized functional space when unable to bridge the structural requirements of disparate functions.

## G. Expanded Formulations for Discrete Diffusion

In Section 3.1, we introduced the core framework of our multimodal discrete diffusion process. Here, we provide the complete mathematical formulations, specifically detailing the closed-form marginal transition of the forward process and the exact posterior derivation for the reverse denoising phase.

### G.1. Forward Process: Closed-Form Marginal Transition

We model the joint corruption of sequence and structure modalities in $\mathcal{P}_{\text{disc}}$ as a discrete-state Markov chain. While the single-step transition is governed by the absorbing matrix $\mathbf{Q}_t = (1 - \beta_t)\mathbf{I} + \beta_t \mathbf{1}_{[\text{MASK}]}$, simulating the entire Markov chain sequentially during training is computationally prohibitive.

To enable efficient optimization, the absorbing formulation allows for a tractable marginal transition directly from the clean data $\mathbf{u}^{(0)}$ to any arbitrary diffusion step $t$. For any individual token $u \in \{s, z\}$, the marginal distribution is defined as:

$$q(u^{(t)} \mid u^{(0)}) = \text{Cat}\left(u^{(t)}; u^{(0)}\bar{\mathbf{Q}}_t\right), \tag{9}$$

where $u$ is represented as a one-hot vector, and $\bar{\mathbf{Q}}_t$ represents the cumulative transition matrix:

$$\bar{\mathbf{Q}}_t = \prod_{i=1}^{t} \mathbf{Q}_i = (1 - \bar{\beta}_t)\mathbf{I} + \bar{\beta}_t \mathbf{1}_{[\text{MASK}]}. \tag{10}$$

Here, $\bar{\beta}_t = 1 - \prod_{i=1}^{t}(1 - \beta_i)$ denotes the cumulative probability of a token being masked at step $t$. This closed-form marginal formulation is essential for the training objective, as it permits the independent and parallel sampling of corrupted sequence and structure states at any arbitrary timestep without iterative simulation.

### G.2. Reverse Process: Exact Posterior Derivation

During the generative phase, the model reconstructs the clean protein $\mathbf{u}^{(0)}$ from the corrupted state $\mathbf{u}^{(t)}$ by reversing the diffusion trajectory, conditioned on the functional semantics $\mathcal{C} = \{\mathbf{C}_{GO}\}$. As defined in the main text, the reverse transition is approximated by marginalizing over the neural network's prediction of the clean state, $\tilde{\mathbf{u}}^{(0)}$.

The step-by-step denoising mechanism requires sampling from the true posterior transition distribution $q(\mathbf{u}^{(t-1)} \mid \mathbf{u}^{(t)}, \tilde{\mathbf{u}}^{(0)})$. Given the Markov property of the forward process, this posterior is derived exactly using Bayes' theorem:

$$q(\mathbf{u}^{(t-1)} \mid \mathbf{u}^{(t)}, \tilde{\mathbf{u}}^{(0)}) = \frac{q(\mathbf{u}^{(t)} \mid \mathbf{u}^{(t-1)})q(\mathbf{u}^{(t-1)} \mid \tilde{\mathbf{u}}^{(0)})}{q(\mathbf{u}^{(t)} \mid \tilde{\mathbf{u}}^{(0)})}. \tag{11}$$

By substituting the single-step transition $q(\mathbf{u}^{(t)} \mid \mathbf{u}^{(t-1)})$ and the marginal transitions into this equation, we obtain a tractable categorical distribution for sampling $\mathbf{u}^{(t-1)}$.

