# OpenReview forum: "Co-Generative De Novo Functional Protein Design"
_ICML.cc/2026/Conference — ICML 2026 regular_

### Official Review · Reviewer_uykT · 2026-03-04

**Soundness:** 4
**Presentation:** 4
**Significance:** 3
**Originality:** 3
**Overall Recommendation:** 5
**Confidence:** 5

**Summary:**

This paper introduces CodeFP, a co-generative protein language model for de novo functional protein design. CodeFP tackles the inherent tension between function and foldability in *de novo* protein design through a co-generative approach. By interleaving the generation of amino acid sequences with structural tokens, the model ensures that functional requirements are physically realizable. The framework's core strength lies in its functional-structural motif retrieval and local supervision, which together sharpen the mapping between abstract functions and precise geometries. Benchmarks confirm that CodeFP consistently outperforms existing baselines, particularly in folding reliability and its ability to generalize to novel, out-of-distribution functional targets. Ablation studies and case analyses further demonstrate the contribution of individual model components.

**Compliance With Llm Reviewing Policy:**

Affirmed.

**Final Justification:**

I am convinced by authors' feedbacks and believe this is a solid work for functional protein design. I suggest a clear acceptance with high confidence.

**Key Questions For Authors:**

1. The experimental results show impressive gains in both pLDDT and F1 scores. However, in cases where the model fails, does it tend to fail on foldability or functionality?

**Limitations:**

Yes

**Strengths And Weaknesses:**

Strengths:

1. The paper is technically grounded in the challenge of balancing functional specificity with structural viability. The idea of a discrete diffusion framework to handle both amino acid sequences and discretized structure tokens is both mathematically and biologically sound for the task.
2. The authors use a robust suite of metrics and provide comprehensive and solid evaluations. Quantitative results in Table 1 and Table 2 show consistent, substantial improvements in both functional F1 and foldability (pLDDT, pTM), while Figures 3–7 visualize diversity, novelty, OOD generalization, and qualitative generation outcomes. With various and up to date baselines, experiment conducted in this paper ensures a solid state-of-the-art performance.
3. The figures (specifically the framework overview and the interleaved decoding process) are high-quality and significantly help in understanding the complex interaction between sequence and structure tokens.
4. The manuscript is well written and organized, making the technical content accessible and reproducible.

Weakness:

1. Figure 2 appears to be a non-diffusion based model at first glance since the input is not sampled from noise but a simply "noised protein". The overall pipeline illustration can be further improved.
2. The diffusion process (forward and backward) is discussed insufficiently compared to the sections of Functional-Structural Retrieval and Local Structure-Function Supervision.
3. If more case studied can be provided (newly updated data, other o.o.d. datasets or fail cases), impact of this work will be more pronounced.

---

> ### Author Rebuttal · Authors · 2026-03-30
>
> # Response to Reviewer uykT
>
> We thank you for the positive evaluation and for recognizing the technical soundness of CodeFP. Below, we address your constructive suggestions regarding presentation, theoretical elaboration, and failure mode analysis.
>
> ---
>
> **1. W1: Pipeline Illustration (Figure 2)**
>
> We agree that the "noised protein" and the lack of explicit forward/backward trajectory visualization in Figure 2 might cause initial confusion regarding the diffusion nature of the model.
>
> Our intention with Figure 2 was to highlight the synergistic interaction between the FSR, LSFS, and the co-generation modules during the training phase. We will update the illustration in the final version to reduce potential ambiguity.
>
> ---
>
> **2. W2: Insufficient Discussion of the Diffusion Process**
>
> We appreciate this constructive feedback. To address this, we have drafted a more comprehensive and detailed mathematical derivation of both the forward absorbing process and the backward denoising trajectory.
>
> This expanded derivation has been temporarily uploaded for review at [Anonymous Link: `https://anonymous.4open.science/r/ICML2026_rebuttal-0B3A/diffusion_writing.png`] and will serve as the basis for improving the presentation in the final version to ensure the co-generation process is thoroughly understood and reproducible.
>
> ---
>
> **3. Q1: In cases where the model fails, does it tend to fail on foldability or functionality?**
>
> We plotted the joint density distribution of pLDDT (foldability) versus F1-Score (functionality) for every individual sample generated in the test set. We have temporarily hosted this plot at [Anonymous Link: `https://anonymous.4open.science/r/ICML2026_rebuttal-0B3A/plddt_vs_f1_density.png`].
>
> The empirical distribution clearly shows that **CodeFP tends to fail on functionality rather than foldability.** We will include this analysis in the final version.
>
> ---
>
> **4. W3: Providing more case studies to make the impact more pronounced.**
>
> We selected two high-pLDDT failure cases from our Out-of-Distribution (OOD) subset. These cases concretely illustrate the hypotheses presented in Section 4.3 regarding generation difficulty tied to GO term depth and semantic distance.
>
> * **Case 1: Insufficient Specificity (High GO Depth Failure)**
>   * **Target Functions:** dTTP/UTP pyrophosphatase activity (GO:0036221, GO:0036218, GO:0047429).
>   * **Predicted Functions:** Pyrophosphatase activity (GO:0016462), nucleotide binding (GO:0000166), ATP binding (GO:0005524), alongside other generic hydrolase terms.
>   * **Generated pLDDT:** 94.47
>   * **Analysis:** While the model correctly identifies the overarching catalytic mechanism (pyrophosphatase activity) and general nucleotide binding, it struggles with precise substrate specificity (predicting ATP instead of dTTP/UTP binding). This highlights that predicting deep, specific GO terms remains challenging, likely due to their reliance on fine-grained structural nuances in the binding pocket.
>
> * **Case 2: Semantic Disconnect (Large Semantic Distance Failure)**
>   * **Target Functions:** nicotinate-nucleotide-dimethylbenzimidazole phosphoribosyltransferase activity (GO:0008939) + nucleotide binding (GO:0000166).
>   * **Predicted Functions:** Catalytic activity (GO:0003824), general binding (GO:0005488), and broad transferase activity (GO:0016740).
>   * **Generated pLDDT:** 93.22
>   * **Analysis:** While producing a stable backbone, the model defaulted to generic GO terms because it failed to physically reconcile the unseen, distant semantic combination (Binding vs. Transferase, distance=8). This validates our finding that larger semantic distances exacerbate compositional generation difficulty.
>
> We will add these detailed qualitative case studies to the Appendix to make the capabilities and current limitations of our work more pronounced.

---

> > ### Author Rebuttal · Reviewer_uykT · 2026-04-01
> >
> > I have carefully read the authors' rebuttal and the additional experimental results provided. The authors have thoughtfully addressed all my concerns, particularly regarding (i) clarification of the diffusion process, (ii) improvement plans for Figure 2, and (iii) the analysis of failure modes. While the paper does not introduce a fundamentally groundbreaking idea or a paradigm-shifting methodology that would warrant the highest score, I still find it to be a solid, well-executed, and meaningful piece of work. The authors demonstrate careful engineering, thorough experimentation, and clear effort in addressing a challenging problem. Overall, I will maintain my original score. Thank authors for their hard work.

---

> > > ### Author Response · Authors · 2026-04-02
> > >
> > > Thank you for your time and favorable comments. We are happy to hear that your concerns have been satisfactorily addressed. Your constructive suggestions have helped us strengthen the paper, and we will incorporate all the discussed improvements into the final version. We are willing to provide additional information if any further questions arise.

---

### Official Review · Reviewer_LJRb · 2026-03-12

**Soundness:** 3
**Presentation:** 3
**Significance:** 4
**Originality:** 4
**Overall Recommendation:** 5
**Confidence:** 4

**Summary:**

The authors introduce CodeFP, a novel framework for _de novo_ functional protein design. The work augments traditional approaches, which either (1) predict sequence directly from function, or (2) sequentially predict structure from function, then sequence from structure; both approaches struggle to simultaneously achieve target functions while maintaining foldability. To address this gap, CodeFP simultaneously encodes local (quantized) structural features and functional motifs, applying cross-attention to integrate the two modalities. The model is trained using a diffusion approach, aiming to maximize the probability of a protein sequence and structure given a target function after denoising. The model is trained and evaluated on 100K+ protein sequences and functional annotations from SwissProt and associated structures from the Protein Data Bank (PDB) and AlphaFoldDB. The authors claim improvements in foldability and functional integrity, even on held-out examples and after generating sequences with low homology to natural variants.

**Compliance With Llm Reviewing Policy:**

Affirmed.

**Key Questions For Authors:**

1. Will the training data be released as part of this publication?
2. How was the example used in Figure 7 selected? There appear to be few, if any, secondary structures in the baseline prediction (panel c). Is this representative of the observed results?
3. Are the evaluation metrics robust to the choice of models used (see Weakness 1 above)?
4. You mention that your training set includes a mixture of PDB and AlphaFoldDB structures. Since the PDB includes experimentally solved structures and AlphaFoldDB includes predicted structures, I would argue that the PDB is more reliable. Have you considered up-weighting examples that are derived from the PDB during training?

**Limitations:**

The authors note a key limitation of this work and many other computational works: the lack of wet-lab validation. This limitation could be elaborated to note that the evaluation metrics used in this study are sensitive to the performance of DeepGO-SE and ESMFold.

**Strengths And Weaknesses:**

**Strengths:**

1. The task addressed by the authors is meaningful and challenging. The ability to confidently design novel, functional proteins would be an important contribution to biotechnology.
2. The authors do a good job contextualizing the task in existing approaches to _de novo_ protein design.
3. The authors leverage an interesting, computational approach to integrate distinct, yet complementary, biological modalities to achieve their design goal.
4. The authors prepare an impressive training set with tens of thousands of proteins with structural and functional annotations.
5. The authors are careful to aggregate the representations of similar structural motifs and GO terms and to address the class imbalance of GO terms. The authors also account for the semantic similarity of GO terms in their evaluation (Figure 5).
6. Figure 7, if representative, is a convincing visualization of the value of CodeFP.

**Weaknesses:**

1. According to section 4.1, **Metrics**, the evaluation metrics rely on other models. Specifically, the functions of generated proteins are assessed by comparing GO terms predicted by DeepGO-SE to target annotations, while foldability is assessed using ESMFold’s pLDDT and pTM scores. The evaluation metrics are sensitive to the quality of the models used in the assessment. If DeepGO-SE and ESMFold have uncertainty in their predicted GO terms and foldability scores, respectively, then this uncertainty will propagate into the evaluation metrics. The authors recognize that wet-lab validation is ideal for assessment. I recognize that this is often impractical. The authors may consider evaluating against additional models to gain a sense of confidence in the metrics.
2. The text in Figures 3-6 is too small and unreadable in print. Please increase the font size in these figures. If space is a constraint, the authors may consider using higher-level figures in the main text, summarizing the existing figures in writing or tables, and moving the existing figures to the appendix in a larger form.
3. Table 2 and Figure 4 could be strengthened by adding uncertainty bounds and error bars, respectively, to highlight the statistical significance of the results.

Overall, this work constitutes an important step towards addressing a challenging problem in biotechnology: the design of novel, functional proteins. For this reason, I would argue that the work is worthy of publication.

---

> ### Author Rebuttal · Authors · 2026-03-30
>
> # Response to Reviewer LJRb
>
> We appreciate your thoughtful evaluation and positive assessment of our work. We address the questions below and will incorporate the suggested improvements in the final version.
>
> ---
>
> **1. W1 & Q3: Robustness of Evaluation Metrics**
>
> We agree that reliance on specific models (DeepGO-SE and ESMFold) may introduce potential metric bias. To rigorously validate the robustness of our results, we conducted additional evaluations using different predictive models. Under these new metrics, we compared our approach against the best-performing baselines from two paradigms: CFP-Gen (one-step) and Pinal (two-step).
>
> **Functional Evaluation via NetGO 4.0 [1]:** We re-evaluated the full test set (8,309 sequences) using NetGO 4.0. We removed predicted GO terms with confidence scores < 0.6, following NetGO 4.0’s definition of high-confidence predictions.
>
> | Model | F1-Micro (↑) | F1-Macro (↑) | AUPR (Macro) (↑) | AUC-ROC (Macro) (↑) |
> | :--- | :--- | :--- | :--- | :--- |
> | Pinal | 0.479 | 0.486 | 0.348 | 0.766 |
> | CFP-Gen | 0.511 | 0.538 | 0.421 | 0.810 |
> | **CodeFP (Ours)** | **0.662** | **0.683** | **0.591** | **0.868** |
>
> Under NetGO 4.0, CodeFP's performance lead actually widens significantly (+0.151 in F1-Micro over CFP-Gen).
>
> **Structural Evaluation via OmegaFold [2]:**
> Due to computational constraints during the rebuttal, we randomly sampled 1,000 sequences from the test set and predicted their structures using OmegaFold.
>
> | Model | pLDDT > 70 (↑) |
> | :--- | :--- |
> | CFP-Gen | 70.10% |
> | Pinal | 75.70% |
> | **CodeFP (Ours)** | **78.50%** |
>
> CodeFP consistently maintains structural superiority under OmegaFold. The strong consistency across different predictive oracles confirms that our evaluation metrics are robust and our reported improvements are empirically significant. We will include these results in the final version.
>
> ---
>
> **2. Q2: Representativeness of the Case Study (Figure 7)**
>
> The example in Figure 7 is drawn from the Out-of-Distribution (OOD) subset, corresponding to a functional combination (Mannitol-1-phosphate 5-dehydrogenase activity + NAD binding) that never co-occurs in the training set.
>
> While the structural collapse in Fig. 7(c) is an extreme case, it reflects a broader trend: proteins generated by the baseline are more prone to structural degradation.
>
> To prove this quantitatively, we analyzed the foldability of the generations within the OOD subset:
>
> | Model | pTM < 0.2 (↓) | pTM < 0.3 (↓) |
> | :--- | :--- | :--- |
> | CFP-Gen | 14.0% | 22.4% |
> | **CodeFP (Ours)** | **1.86%** | **9.57%** |
>
> The CFP-Gen (Baseline) exhibits significant structural degradation: 14.0% of its OOD generations yield pTM < 0.2, compared to only 1.86% for CodeFP.
>
> ---
>
> **3. Q4: Up-weighting PDB over AlphaFoldDB**
>
> This is a sound suggestion. Assigning higher sample weights to experimentally resolved structures (PDB) could mitigate the learning of artifacts present in AlphaFoldDB predictions. However, a full training cycle requires 200 A800 GPU hours, and comprehensive evaluation takes ~138 GPU hours plus 6h CPU time.
>
> Given these computational constraints, we could not complete a full retraining during the rebuttal. We will explore this weighting hyperparameter and include the findings in the final version if empirical gains are observed.
>
> ---
>
> **4. W2, W3 & Q1: Presentation, Statistical Reporting, and Data Release**
>
> We appreciate these suggestions and will address them in the final version.
>
> * **Presentation:** We will increase the font size in Figures 3–6 to improve readability in print.
> * **Statistical Reporting:** Due to time constraints during the rebuttal, we have not yet added uncertainty bounds and error bars; we will include them in the final version.
> * **Data Release:** We will publicly release the training dataset, including the processed FSR database, along with the CodeFP model weights upon publication.
>
> # Reference
>
> [1] Yan et al., "GOAnnotator: accurate protein function annotation using automatically retrieved literature." Bioinformatics (ISMB), 2025.
>
> [2] Wu et al., "High-resolution de novo structure prediction from primary sequence." bioRxiv, 2022.

---

> > ### Author Rebuttal · Reviewer_LJRb · 2026-04-02
> >
> > Thank you for incorporating my feedback and answering my questions.
> >
> > Regarding **W1 & Q3**, your additional validator model experiments increase my confidence that CodeFP’s superior performance is robust to metric bias.
> >
> > Regarding **Q2**, the table you provide in the rebuttal is convincing. In addition to including these results, please clearly state that Fig. 7(c) is an extreme case in your paper. Alternatively, you may consider using a more representative example in your comparison.
> >
> > Regarding **Q4**, given the computational expense of training and evaluating the model, I acknowledge that retraining to up-weight PDB examples may be impractical at this time. My question may be taken as a consideration for future model development.
> >
> > Regarding **W2, W3 & Q1**, I appreciate your plans to improve the readability and statistical rigor of this work. Your promise to release the training dataset and model weights adds to the impact of your work. CodeFP will be a valuable resource for the community.
> >
> > In conclusion, your rebuttal addresses my main concerns and supports my recommendation to accept the paper. Thank you for sharing this interesting work.

---

> > > ### Author Response · Authors · 2026-04-03
> > >
> > > Thank you for your positive feedback and your support for accepting our paper. We are glad that our additional experiments and responses have fully resolved your concerns. Your insightful comments have helped us strengthen the paper significantly. We are willing to provide additional information should any further questions arise.

---

### Official Review · Reviewer_TDC8 · 2026-03-13

**Soundness:** 3
**Presentation:** 2
**Significance:** 3
**Originality:** 2
**Overall Recommendation:** 3
**Confidence:** 5

**Summary:**

CodeFP addresses the limitations of one-step and two-step protein generation: the 'one-to-many' ambiguity of the former and the structural instability (disordered coils) of the latter. By adopting a Co-generation strategy, the model simultaneously decodes sequence and structure tokens to ensure physical realizability. To mitigate information loss from structure quantization, FSR injects motif-level priors via cross-attention. Additionally, LSFS utilizes IPS-based indexing and auxiliary classification loss to reduce latent ambiguity and align the generative distribution with target GO labels. This framework effectively grounds functional semantics in structural motifs, enabling precise de novo functional protein design.

**Compliance With Llm Reviewing Policy:**

Affirmed.

**Key Questions For Authors:**

Q1. The w/o FSR & LSFS variant seems to be the closest backbone-only baseline to the full model. Could the authors discuss the final gains more explicitly relative to this variant, so that the contribution of FSR/LSFS can be separated more clearly from that of the DPLM-2 backbone?

Q2. Please revise the manuscript for textual and numerical consistency. In particular, the typo in Section 4.5, the phrase “traning set frequency,” the “Metallo-petidase” entry, and the mismatch between the foldability paragraph and Table 2 should be corrected.

Q3. As written, the FSR retrieval database appears to be built from training data only. Please state this explicitly and clarify whether any validation/test proteins or annotations are excluded from motif construction.

Q4. Were any evaluation sequences already included in DPLM-2 pretraining?

**Limitations:**

Yes

**Strengths And Weaknesses:**

S1. The paper logically defines the limitations of one-step (search space ambiguity) and two-step (structural collapse) methods, providing a persuasive rationale for a co-generation approach.

S2. FSR effectively grounds functional semantics into structural motifs, while LSFS mitigates structure token ambiguity, creating a robust modular synergy for precise design.

S3. The evaluation is thorough, with OOD analyses and ablation studies that provide strong empirical confidence in the model’s generalization and functional integration.

W1. Clerical Errors and Data Inconsistencies
- The title of Section 4.5 is misspelled as "Abaltion Study."
- The term "traning set frequency" appears in the main text.
- In Table 5, "Metallo-petidase" should be corrected to "Metallo-peptidase."
- Section 4.2 cites pLDDT as 80.65% vs. 75.52% as if both are compared to Pinal, but Table 2 reveals 75.52% actually belongs to CFP-Gen.

W2. The paper is strongly motivated by the dichotomy between one-step (sequence-first) and two-step (structure-first) generation. However, since CodeFP is technically a co-generative (or co-design) extension built on top of DPLM-2, this framing is not perfectly aligned with the actual technical implementation. This gives rise to two related concerns:

Experimental: Since the model updates only the cross-attention modules and the LSFS head while freezing the DPLM-2 backbone, the w/o FSR & LSFS ablation appears to be the most direct baseline for assessing the value of the proposed modules. However, this comparison is not sufficiently foregrounded in the discussion of performance gains. As a result, it remains somewhat unclear how much of the improvement should be attributed to the strength of the pre-trained co-generative backbone versus the newly introduced FSR/LSFS mechanisms.

Conceptual: While the one-step vs. two-step distinction is effective for motivating the problem, it may not be the most informative lens for isolating the paper’s actual novelty. Since models that jointly model sequence and structure are increasingly understood under a broader co-design or co-generation paradigm, the manuscript could benefit from positioning CodeFP more explicitly relative to such references. Clarifying whether the reported gains stem from addressing broad paradigm-level limitations or from introducing function-aware extensions on top of a strong co-design backbone would strengthen the paper.

---

> ### Author Rebuttal · Authors · 2026-03-30
>
> # Response to Reviewer TDC8
>
> We appreciate your careful reading and constructive feedback. We address the concerns below and will incorporate the corresponding clarifications and corrections in the final version.
>
> ---
>
> **1. W1 & Q2: Clerical Errors and Data Inconsistencies**
>
> Thank you for pointing this out. These clerical errors and inconsistencies will be corrected in the final version.
>
> ---
>
> **2. W2 & Q1: Isolating module contributions**
>
> We clarify that the `w/o FSR & LSFS` variant is a fully trained co-generation model guided purely by one-hot GO embeddings. It combines a frozen DPLM-2 backbone with a trainable gated network (following CFP-Gen) and a cross-attention module that jointly inject the GO embeddings. Therefore, this serves as the exact baseline to demonstrate the original functional design capability of the co-generation paradigm. Additionally, we reiterate the definition of the `two-step generation` variant: it shares the same architecture as `w/o FSR & LSFS` but is trained with a structure-only objective. During inference, it generates the structure first, then derives the sequence via inverse folding.
>
> To clarify the source of the performance gains and better contextualize our Ablation Study (Section 4.5), we revisit these results below:
>
> | Model Variant / Paradigm | F1-Micro (↑) | F1-Macro (↑) | MMD (↓) | pLDDT > 70 (↑) | pTM > 0.5 (↑) |
> | :--- | :--- | :--- | :--- | :--- | :--- |
> | CFP-Gen (one-step) | 0.429 | 0.370 | 0.112 | 75.52% | 72.30% |
> | Pinal (two-step)| 0.452 | 0.369 | 0.223 | 74.22% | 82.22% |
> | **`two-step generation`** | 0.414 | 0.285 | 0.282 | 52.24% | 59.54% |
> | **`w/o FSR & LSFS`**| 0.465 | 0.400 | 0.192 | 71.71% | 71.98% |
> | **CodeFP** | **0.496** | **0.446** | **0.106** | **80.65%** | **83.48%** |
>
> **Decomposing the Gains:**
> 1.  **Paradigm Gain:** The co-generation model (`w/o FSR & LSFS`) outperforms one-step generation (CFP-Gen, +0.036 F1-Micro). In contrast, the (`two-step generation`) variant leads to severe structural degradation (pLDDT > 70 drops to 52.24%), highlighting the limitation of two-step generation.
> 2.  **FSR & LSFS Module Gain:** When integrating our proposed FSR and LSFS modules into the co-generation baseline, the model achieves a massive leap in both functionality (F1-Micro +0.031) and foldability (pLDDT > 70 improves by +8.94%).
>
> Experimentally, this cleanly separates paradigm and module contributions: co-generation provides a stronger foundation than one-step/two-step approaches, while FSR and LSFS are key to achieving state-of-the-art joint performance. We will clarify this decomposition more explicitly in Section 4.5 in the final version.
>
> Conceptually, our novelty is twofold. First, it lies in the insight that the inherent bottleneck of functional protein design, achieving simultaneous foldability and functionality, can be fundamentally resolved through joint sequence-structure modeling. Second, directly applying the co-generation paradigm to functional design introduces non-trivial challenges, specifically information loss from structure quantization and training ambiguity. Our FSR and LSFS modules are meticulously designed to overcome these exact adaptation bottlenecks, thereby unlocking the substantial performance gains reported.
>
> ---
>
> **3. Q3: FSR Retrieval Database Construction**
>
> We confirm that the FSR retrieval database was constructed *exclusively* using the training set. We will state this explicitly in Section 3.2 to address any concerns regarding data leakage.
>
> ---
>
> **4. Q4: Evaluation sequences in DPLM-2 pretraining**
>
> Among the 8,309 test sequences, 3,907 sequences were present in the DPLM-2 pretraining corpus. However, this does not constitute data leakage for our specific task.
>
> DPLM-2 pretraining optimizes for the unconditional distribution $P(\text{protein})$. In contrast, *de novo* functional protein design evaluates the conditional generative capability $P(\text{protein}|\text{function})$. Exposure to a sequence during unconditional pretraining does not equip the model to learn the complex mapping between function and their specific sequence-structure instantiations.
>
> To prove that our model's performance does not rely on memorized unconditional distributions, we filtered out the 3,907 overlapping sequences and re-evaluated the models on the remaining 4,402 test proteins:
>
> | Model | F1-Micro (↑) | F1-Macro (↑) | AUPR (Macro) (↑) | AUC-ROC (Macro) (↑) | pLDDT > 70 (↑) | pTM > 0.5 (↑) |
> | :--- | :--- | :--- | :--- | :--- | :--- | :--- |
> | Pinal | 0.448 | 0.377 | 0.237 | 0.668 | 71.51% | **79.64%** |
> | CFP-Gen | 0.448 | 0.409 | 0.285 | 0.697 | 73.28% | 68.94% |
> | **CodeFP (Ours)** | **0.490** | **0.452** | **0.328** | **0.727** | **78.28%** | 79.05% |
>
> CodeFP maintains its substantial lead in both functional consistency (F1-Micro +0.042 over baselines) and structural realizability (pLDDT > 70 +5.00% over CFP-Gen). We will include this evaluation in the final version.

---

### Official Review · Reviewer_g4Q6 · 2026-03-15

**Soundness:** 3
**Presentation:** 3
**Significance:** 3
**Originality:** 2
**Overall Recommendation:** 5
**Confidence:** 4

**Summary:**

This paper introduces CodeFP, a co-generative protein language model framework for de novo functional protein design. The method simultaneously generates amino acid tokens and discrete structure tokens, conditioned on Gene Ontology (GO) molecular function terms. CodeFP is built on the DPLM-2 backbone and introduces two plug-and-play modules: Functional-Structural Retrieval (FSR), which retrieves and aggregates structural motifs associated with GO terms and injects them via cross-attention, and Local Structure-Function Supervision (LSFS), which provides auxiliary classification supervision on continuous hidden states to mitigate training ambiguity from structure tokenization.
Experiments on a GO-conditioned protein design benchmark show consistent improvements over the strong one-step and two-step baselines. OOD generalization experiments on unseen and hypothetical functional combinations indicates the generalizability of the method. Ablation studies cleanly decompose the contributions of FSR, and LSFS.

**Compliance With Llm Reviewing Policy:**

Affirmed.

**Final Justification:**

The authors have adequately addressed my main concerns in the rebuttal. I think this is a meaningful work with a clear technical contribution to functional protein design. The experiments are solid, and the empirical results are strong enough to support the paper’s claims.

**Key Questions For Authors:**

1. The current design freezes the entire DPLM-2 backbone and only trains the newly added modules. Would training the DPLM-2 parameters together (such as LoRA) bring greater gains?
2. Can the proposed method support classifier guidance (using the trained LSFS prediction head) or classifier-free guidance at inference time? Will this introduce more benefits?

**Limitations:**

There is no limitation discussion in the paper.

**Strengths And Weaknesses:**

## Strengths
1. CodeFP introduces two plug-and-play modules (FSR, LSFS) to the pre-trained protein language models, enabling the functional design capability. During training, the parameters of the pre-trained models do not need to be trained, making the proposed method easily applicable to existing models.
2. CodeFP achieves the best performance across F1-Micro, F1-Macro, AUPR, AUC-ROC, MRR, pLDDT > 70, and pTM > 0.5 among one-step and two-step baselines.
3. This paper shows clear ablation study, which decomposes contributions of the proposed modules: FSR improves foldability (+10 percentage points on pLDDT > 70), and LSFS primarily aids distributional alignment.
4. The evaluation on unseen natural combinations (76 held-out GO combinations) and hypothetical combinations (119 synthetically constructed pairs) goes beyond standard benchmark evaluation, demonstrating the generalizability of the method.

## Weaknesses

1. Lack of ESM3 baseline. ESM3 supports function keyword track as input, enabling native function-conditioned generation.
2. The author should add a baseline settings without any architecture changes. Specifically, for the input GO term, extract the local structure backbone coordinates and discretize to structure token, which can be viewed as structural motifs. Then run the motif-scaffolding inference, as supported by DPLM-2. This would clearly indicate the original functional design capability of the DPLM-2 and demonstrate the contribution and benefit of the proposed method.
3. The author leverage the [CLS] token embeddings as the representation of FSR. However,  the [CLS] token seems just a special token that was not explicitly trained to summarize the semantics of the entire input sequence. (I don't find the relevant explanation in the DPLM-2 paper). So there is no guarantee that [CLS] provides a meaningful representation for FSR. Is there any better design choice? (e.g., average the embeddings of struct tokens)
4. I have some concerns about the training cost. The paper reports a total of 1.59B parameters, of which only 650M come from the frozen DPLM-2 backbone. This means ~940M new parameters are introduced, but why only introducing cross-attention modules leads to so much parameters? Considering nearly the double model size, will this introduce much more training cost? Additionally, the preprocessing pipeline for the FSR retrieval database (InterProScan domain localization, LFQ encoding, [CLS] extraction, aggregation for all GO-protein pairs) may introduce substantial offline computational cost that is not discussed.

---

> ### Author Rebuttal · Authors · 2026-03-30
>
> # Response to Reviewer g4Q6
>
> Thank you for the constructive feedback. We have addressed your concerns in detail below, and these clarifications will be incorporated into the final version.
>
> ---
>
> **1. W1 Lack of ESM3 baseline.**
>
> We evaluated the open-weights ESM3. Because ESM3 requires explicit sequence positions for InterPro (IPR) tracks, we mapped GO terms to IPR and provided ground-truth motif positions.
>
> | Model | F1-Micro (↑) | F1-Macro (↑) | AUPR (Macro) (↑) | AUC-ROC (Macro) (↑) |
> | :--- | :--- | :--- | :--- | :--- |
> | ESM3 | 0.221 | 0.044 | 0.040 | 0.508 |
> | **CodeFP** | **0.496** | **0.446** | **0.321** | **0.724** |
>
> CodeFP outperforms ESM3 across all functional metrics.
>
> ---
>
> **2. W2 Baseline without architecture changes (DPLM-2 Motif-Scaffolding).**
>
> We implemented the suggested DPLM-2 motif-scaffolding baseline. For a random sample of ~400 test proteins, we extracted ground-truth motif structure tokens corresponding to each GO term (capped at a maximum length of 30 per term to manage overly long motifs), masked the remaining amino acid and structure tokens, and evaluated the scaffolding capability:
>
> | Model | F1-Micro (↑) | F1-Macro (↑) | AUPR (Macro) (↑) | AUC-ROC (Macro) (↑) |
> | :--- | :--- | :--- | :--- | :--- |
> | DPLM-2 (Motif-Scaffolding) | 0.449 | 0.453 | 0.339 | 0.758 |
> | **CodeFP** | **0.520** | **0.550** | **0.442** | **0.794** |
>
> Even though the baseline is provided with ground-truth structure, CodeFP still achieves better functional consistency. For a clearer view of DPLM-2's original functional capabilities, please refer to our ablation analysis in the response to Reviewer TDC8 (Point 2).
>
> ---
>
> **3. W3 Validity of the [CLS] token for FSR representation.**
>
> Average pooling of structure tokens is a viable alternative that we plan to explore. However, our current [CLS] embeddings already effectively capture structural semantics. By grouping the [CLS] embeddings according to their corresponding GO terms, we observed highly distinct clustering. This is confirmed by significant statistical test results: PERMANOVA ($F=127.53$, $p=0.001$), ANOSIM ($R=0.704$, $p=0.001$), and $\eta^2=0.573$. These metrics demonstrate that the [CLS] token successfully encodes the structural features of functional motifs.
>
> ---
>
> ---
>
> **4. W4 Training cost, parameter count, and offline FSR overhead.**
>
> We clarify a typo: the total parameter count for CodeFP is ~1.87B. The breakdown is as follows:
>
> * **Frozen DPLM-2 backbone:** ~670M
> * **A gated network (following CFP-Gen) and a cross-attention module, both used to inject one-hot GO embeddings:** ~930M
> * **FSR module:** ~270M
> * **Total:** ~1.87B
>
> **Training Cost:**
> The validation loss converges stably, and we use the final checkpoint at 110K steps for all evaluations (~48 hours on 4 A800 GPUs).
>
> **Offline FSR Cost:**
> Constructing the FSR database is a one-time offline cost. Sequentially, InterProScan takes ~15.6 s/protein on an Intel Xeon 8374C CPU (trivially parallelizable). GPU steps (LFQ encoding, [CLS] extraction) achieve high throughputs of ~5,800 and ~3,900 residues/s on a single A800 (<10GB VRAM). Finally, GO-protein aggregation involves only lightweight CPU vector operations with negligible overhead.
>
> ---
>
> **5. Q1: Would training the DPLM-2 parameters together (such as LoRA) bring greater gains?**
>
> We explored fine-tuning the DPLM-2 backbone with LoRA for 30K steps due to time constraints. However, evaluation on a 1K test subset showed a slight performance drop:
>
> | Setup | F1-Micro (↑) | F1-Macro (↑) | AUPR (Macro) (↑) | AUC-ROC (Macro) (↑) |
> | :--- | :--- | :--- | :--- | :--- |
> | CodeFP (Frozen Backbone) | **0.490** | **0.484** | **0.365** | **0.751** |
> | CodeFP (LoRA Backbone) | 0.483 | 0.476 | 0.354 | 0.745 |
>
> While it may reflect the model adapting pre-trained representations rather than learning new functional mappings, the exact cause remains unclear and may also depend on training duration.
>
>
> ---
>
> **6. Q2: Support for classifier guidance or classifier-free guidance (CFG) at inference.**
>
> * **Classifier Guidance (CG):** CodeFP cannot support standard CG during diffusion. The LSFS head requires fully formed *local structures*, which cannot be accurately isolated mid-generation.
> * **Classifier-Free Guidance (CFG):** CodeFP natively supports CFG (trained with 10% condition dropout). Using a CFG scale of 1.5 ($\text{logits} = \text{uncond} + 1.5 \times (\text{cond} - \text{uncond})$) on 800 test proteins yields:
>
> | Inference Mode | F1-Micro (↑) | F1-Macro (↑) | pLDDT > 70 (↑) | pTM > 0.5 (↑) |
> | :--- | :--- | :--- | :--- | :--- |
> | Standard (Scale = 1.0) | 0.497 | 0.492 | **83.75%** | 84.50% |
> | CFG (Scale = 1.5) | **0.507** | **0.501** | 81.12% | **86.12%** |
>
> CFG enforces stronger functional alignment (improved F1 and pTM) with a minimal trade-off in local structural confidence (pLDDT).

---

> > ### Author Rebuttal · Reviewer_g4Q6 · 2026-04-03
> >
> > Thank you for the detailed rebuttal and the additional experiments. I have read the response carefully, and I appreciate the authors’ effort in addressing my previous concerns.
> >
> > I still have one remaining question regarding the parameter count. In particular, I find it somewhat surprising that the gated network and cross-attention modules together (\~930M) contain more parameters than the original DPLM-2 backbone (\~670M). Could the authors clarify how the parameters in these modules are composed, and why their total size is so large?
> >
> > Relatedly, if a larger pretrained protein language model were used as the backbone (e.g., ESM-3 1.4B), would the number of trainable parameters in the added modules also increase accordingly? A brief explanation of this scaling behavior would help me better understand the efficiency and design trade-offs of the proposed method.

---

> > > ### Author Response · Authors · 2026-04-03
> > >
> > > Thank you for your follow-up questions regarding the parameter composition and scaling behavior.
> > >
> > > **1. Composition of the ~930M Parameters**
> > >
> > > The \~930M parameters consist of the gated network (\~660M) and the cross-attention module (~270M, sharing the same architecture as FSR).
> > >
> > > For a hidden size $H$, the gated network closely follows the architecture of CFP-Gen and primarily uses a linear projection of $H \rightarrow 12H$. The factor of 12 comes from: (i) independent modulation of sequence and structure (scale, shift, gate $\rightarrow 3 + 3 = 6$), and (ii) applying this modulation twice per layer ($6 \times 2 = 12$). This results in $12H^2 + 12H$ parameters per layer. With $H = 1280$ and $L = 33$, this accounts for the vast majority of the ~660M parameters.
> > >
> > > The cross-attention module contributes approximately $5LH^2$ (~270M), comprising the functional projection ($H^2+H$), layer normalization ($2H$), and multi-head attention ($4H^2+4H$).
> > >
> > > The overall ~1.87B parameter count of CodeFP is distributed approximately as follows: frozen DPLM-2 backbone (36%), gated network (35%), cross-attention module (14%), and FSR module (14%).
> > >
> > > **2. Scaling Behavior**
> > >
> > > If a larger backbone like ESM-3 is used, the trainable parameters will scale proportionally to $O(L \cdot H^2)$ based on the specific constants of each module: $\approx 12LH^2$ for the gated network, $\approx 5LH^2$ for the cross-attention module, and $\approx 4LH^2 + L \cdot d_{\text{motif}}H$ for the FSR module.
> > >
> > > We will include this detailed parameter breakdown and scaling analysis in the appendix of the final version. We are willing to provide additional information if any further questions arise.

---

### Decision · Program_Chairs · 2026-04-30

**Decision:**

Accept (regular)

**Comment:**

The reviewers overall agree that this is a good paper, pointing out the solid execution of the work and the strong empirical results. Moreover, the authors provided extensive additional analyses and experiments during the rebuttal phase and were able to address all concerns.

My own assessment of the paper agrees with the reviewers' assessments of the manuscript, and I believe that this is a meaningful contribution to ICML. Therefore, I recommend acceptance of the paper.